# The PripA-TbcrA complex-centered Rab GAP cascade facilitates macropinosome maturation in *Dictyostelium*

Hui Tu[1,2], Zhimeng Wang[1,2], Ye Yuan[1,2], Xilin Miao[1,2], Dong Li[1,3], Hu Guo[1,2], Yihong Yang[1] & Huaqing Cai [1,2✉]

Macropinocytosis, an evolutionarily conserved mechanism mediating nonspecific bulk uptake of extracellular fluid, has been ascribed diverse functions. How nascent macropinosomes mature after internalization remains largely unknown. By searching for proteins that localize on macropinosomes during the Rab5-to-Rab7 transition stage in *Dictyostelium*, we uncover a complex composed of two proteins, which we name PripA and TbcrA. We show that the Rab5-to-Rab7 conversion involves fusion of Rab5-marked early macropinosomes with Rab7-marked late macropinosomes. PripA links the two membrane compartments by interacting with PI(3,4)P$_2$ and Rab7. In addition, PripA recruits TbcrA, which acts as a GAP, to turn off Rab5. Thus, the conversion to Rab7 is linked to inactivation of the upstream Rab5. Consistently, disruption of either *pripA* or *tbcrA* impairs Rab5 inactivation and macropinocytic cargo processing. Therefore, the PripA-TbcrA complex is the central component of a Rab GAP cascade that facilitates programmed Rab switch and efficient cargo trafficking during macropinosome maturation.

[1] National Laboratory of Biomacromolecules, Institute of Biophysics, Chinese Academy of Sciences, 100101 Beijing, China. [2] College of Life Sciences, University of Chinese Academy of Sciences, 100049 Beijing, China. [3] School of Life Sciences and Medicine, University of Science and Technology of China, 230027 Hefei, China. ✉email: huaqingcai@ibp.ac.cn

Macropinocytosis is an evolutionarily conserved mechanism for nonspecific bulk uptake and internalization of extracellular fluid[1]. The process has been observed in a variety of cell types. In the social amoeba *Dictyostelium discoideum*, macropinocytosis is the major route by which axenically grown cells obtain nutrients[2]. In mammals, macropinocytosis plays an important role in antigen sampling, pathogen infiltration, cell migration, and regulation of synaptic activity[3–6]. Recently, macropinocytosis was shown to facilitate tumor cell survival and proliferation in the nutrient-poor tumor microenvironment by engulfing extracellular proteins and fatty acids[7,8]. Recognition of the diverse importance of macropinocytosis in physiology and disease has prompted studies to elucidate the molecular mechanisms underlying this specialized form of endocytosis.

Macropinosomes are much larger in size than endosomes derived from microendocytic pathways, such as clathrin-mediated endocytosis. The capture and processing of large volumes of extracellular fluid poses unique challenges for the cell. Macropinosome initiation requires coordinated three-dimensional manipulation of the plasma membrane, which is driven by actin polymerization to form distinctly shaped membrane protrusions[9–11]. Closure of the protrusions and subsequent membrane fission result in the formation of micrometer-sized aqueous-filled vesicles. Nascent macropinosomes undergo a complex series of maturation steps that reduce the surface area and volume[12,13]. Next, they either recycle to the cell surface or traffic through the endolysosomal system, where their contents may be digested and transported into the cytoplasm for anabolic metabolism[4,14].

Although much has been learned about the membrane lipid and cytoskeleton requirements of macropinosome formation, the molecular machinery regulating the subsequent maturation steps remains to be determined. Macropinosome maturation may share common elements with other endocytic pathways. For clathrin-mediated endocytosis, Rab GTPases are crucial regulators of endosome maturation[15–17]. Early endosomes accumulate active Rab5 via a positive feedback loop consisting of the Rab5-specific guanine nucleotide exchange factor (GEF) Rabex5 and effector protein Rabaptin5[18,19]. As early endosomes convert into late endosomes, Rab5 together with PI3P recruits a protein complex containing Mon1 and Ccz1[20,21]. Mon1 displaces Rabex-5, whereas the Mon1-Ccz1 complex acts as a GEF for Rab7, leading to Rab7 activation[21–23]. How Rab5 is subsequently turned off to complete the cycle is not completely understood. This step likely requires Rab5-specific GTPase-activating proteins (GAPs). In mammalian cells, RabGAP-5 regulates Rab5 inactivation during endocytosis[24], but how it is temporally regulated is not known. Inactivation of the yeast Rab5 Vps21 requires yeast Rab7 Ypt7, the GAP Msb3, and endosome-vacuole fusion[25,26]. It is not yet clear how Msb3 specifically targets Vps21 upon arrival at the vacuole. Similarly, although studies of phagocytosis in *C. elegans* suggest that the class III PI3K Vps34 and PI3P recruit the GAP TBC-2 to inactivate Rab5[27,28], additional factors likely regulate the timing of their action because Vps34 and PI3P are also part of a positive feedback loop for the assembly of Rab5 effector complexes[29,30].

Previous studies have suggested that macropinosomes acquire Rab5 and Rab7 in a similar sequential manner after scission from the plasma membrane[31]. Whether a timely controlled cascade of events regulates the Rab5-to-Rab7 conversion and subsequent Rab5 inactivation on macropinosomes remains to be determined. Considering the increased demand for membrane and fluid processing during macropinocytosis, pathway-specific trafficking steps or regulatory machinery are likely involved.

In the present study, we investigated the macropinosome maturation process in *Dictyostelium*. The standard laboratory strains have high fluid uptake when grown in nutrient medium, more than 90% of which is due to macropinocytosis, making them an excellent model for such a study[2,32]. By performing time-lapse imaging of macropinosome maturation during the Rab5-to-Rab7 transition stage, we unexpectedly found that conversion of Rab5 to Rab7 involves fusion of Rab5-marked early macropinosomes with Rab7-marked late macropinosomes. Furthermore, by screening for proteins with specific localization during this process, we uncovered a protein complex composed of a protein containing a pleckstrin homology (PH) domain and a protein containing a Rab GAP domain and regulator of chromosome condensation 1 (RCC1) domains, which we named PripA (Phospholipid and Rab interacting protein A, DDB_G0275795) and TbcrA (TBC- and RCC1-domain-containing protein A, DDB_G0282575), respectively. Our results indicate that the PripA-TbcrA complex promotes Rab5-to-Rab7 conversion and macropinosome maturation.

## Results

**Macropinosome fusion mediates Rab5-to-Rab7 conversion**. To begin dissecting the macropinosome maturation process in *Dictyostelium*, we monitored the transition from Rab5 to Rab7, which is thought to signify the conversion from early to late endosomal compartments in various endocytic pathways. To follow macropinosomes at different stages after internalization, we incubated cells with 70 kDa TRITC-dextran (TD) or DQ-BSA. TD is a fluid-phase tracer considered specific for the macropinocytic pathway[33,34]; the TD signal was relatively weak in newly formed macropinosomes and increased gradually as macropinosomes shrunk and moved away from the plasma membrane (Supplementary Fig. S1a). DQ-BSA is a self-quenched albumin that only becomes fluorescent upon degradation, and it can be expected to mark macropinosomes that have acquired an acidic and proteolytically active environment[32,35].

First, we examined the localization of Rab5 and Rab7. The *Dictyostelium* genome encodes two Rab5 proteins, Rab5A and Rab5B, which exhibited similar localization when expressed as GFP fusion proteins from extrachromosomal vectors (Supplementary Fig. S1b). To avoid the potential effect of Rab5 overexpression on macropinosome maturation, as has been observed during endosome maturation[36,37], we generated cells in which GFP-Rab5A was expressed from an expression cassette integrated into the genome as a stable single copy via restriction enzyme-mediated integration (REMI)[38–40]. GFP-Rab5A$^{\text{REMI}}$ localized to the plasma membrane and vesicular structures near the nucleus and was enriched on newly formed macropinosomes containing a weak TD signal and minimal DQ-BSA signal (Fig. 1a, b). The expression of GFP-Rab5A$^{\text{REMI}}$ did not affect the TD uptake or DQ-BSA degradation activity, and it did not alter the size of TD-labeled macropinocytic vesicles (Supplementary Fig. S1e–g). The *Dictyostelium* genome also encodes two Rab7 proteins, Rab7A and Rab7B, which localized to cytoplasmic vesicles when expressed from extrachromosomal vectors, though GFP-Rab7B exhibited a weaker membrane association (Supplementary Fig. S1c). We confirmed that GFP-Rab7A-positive vesicles were late macropinosomes, as they contained bright luminal signals for TD and DQ-BSA (Fig. 1c, d). The expression of GFP-Rab7A did not affect TD uptake or DQ-BSA degradation (Supplementary Fig. S1h, i).

Next, we visualized the dynamics of Rab5A and Rab7A simultaneously during macropinocytosis by expressing RFP-Rab7A in GFP-Rab5A$^{\text{REMI}}$ cells. Following newly formed

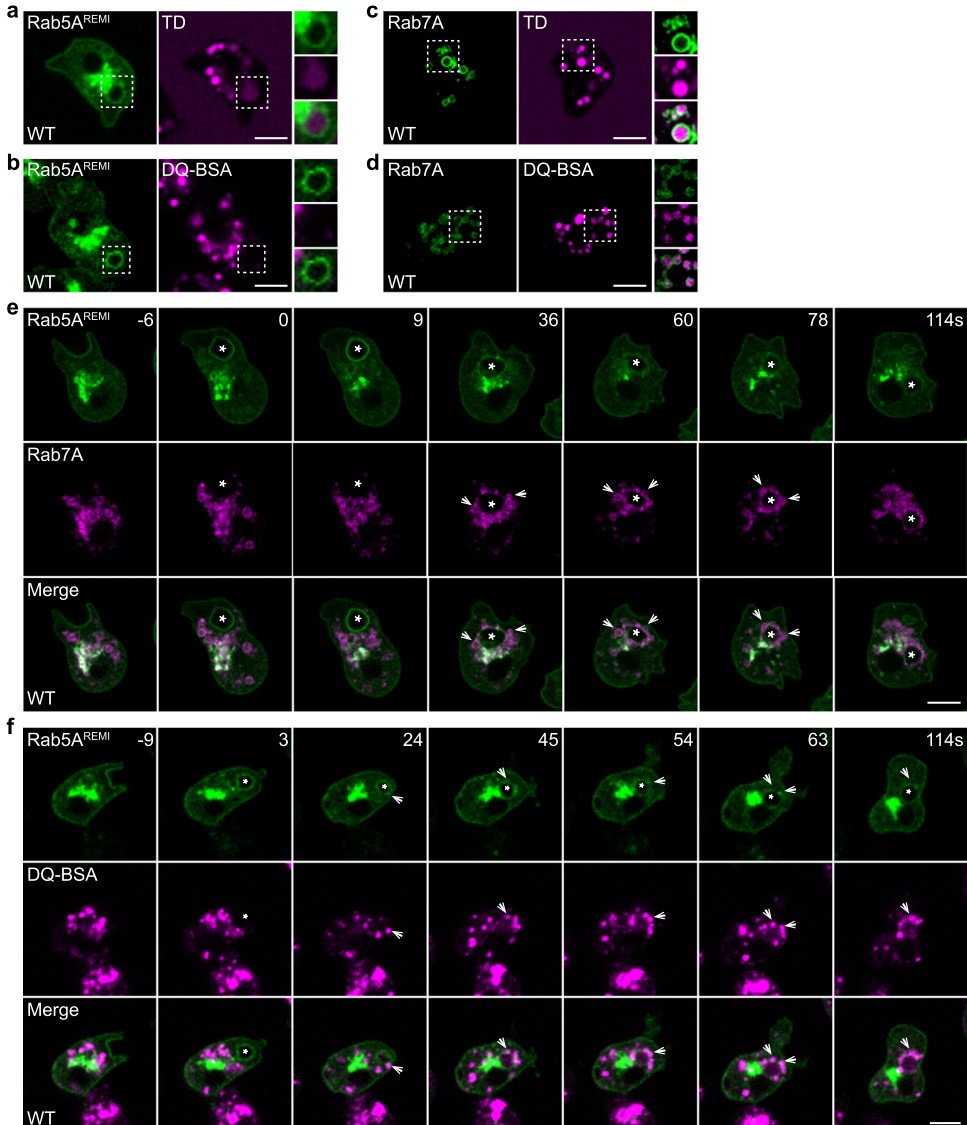

**Fig. 1 Vesicle fusion mediates Rab5-to-Rab7 conversion during macropinosome maturation. a–d** The uptake of TRITC-dextran (TD) or DQ-BSA in WT cells expressing GFP-Rab5A[REMI] or GFP-Rab7A. Images were acquired after a 60-min incubation. Magnified images of the areas indicated by dashed boxes are shown at the right. **e** Time-lapse images of macropinocytosis in WT cells expressing GFP-Rab5A[REMI] and RFP-Rab7A. The asterisks mark a newly formed macropinosome initially decorated by GFP-Rab5A[REMI]. The arrows mark RFP-Rab7A-labeled late macropinosomes that surrounded and fused with the newly formed macropinosome. The time of macropinocytic cup closure is defined as "0 s". **f** Time-lapse images of GFP-Rab5A[REMI]/WT cells pre-incubated with DQ-BSA for 60 min. Images were acquired after DQ-BSA was washed out. The arrows mark preformed macropinosomes containing bright DQ-BSA signal. The asterisk marks a newly generated macropinosome, which gradually obtained DQ-BSA signal through vesicle fusion. Scale bar = 5 μm.

macropinosomes, we observed rapid conversion of the macropinosomal membrane perimeter from Rab5-positive to Rab7-positive (Fig. 1e). Similar results were observed when both GFP-Rab5A and RFP-Rab7A were expressed from integrated expression cassettes (Supplementary Fig. S2a, b). Intriguingly, before Rab5A was lost and Rab7A became concentrated on the membrane, Rab5A-marked macropinosomes appeared to be surrounded by smaller Rab7A-marked macropinosomes starting approximately 25–30 s after cup closure (Fig. 1e, Supplementary Fig. S2a, Supplementary Movie 1). The close apposition of two different macropinocytic compartments prompted us to speculate that, in addition to recruiting regulatory factors from the cytosol, fusion of early macropinosomes with late macropinosomes may contribute to the transition of Rab5 to Rab7.

To verify that vesicle fusion occurs between early and late macropinosomes during Rab5-to-Rab7 conversion, we performed a pulse-chase experiment. GFP-Rab5A[REMI] cells were incubated with DQ-BSA. After 1 h, DQ-BSA was removed, and the cells washed, placed in fresh medium, and imaged. Because DQ-BSA was removed from the extracellular medium, newly formed macropinosomes should contain no DQ-BSA and be easily distinguished from intracellular vesicles containing bright DQ-BSA signal, which were Rab7A-positive macropinosomes. We observed that, as a newly formed Rab5A-marked macropinosome entered the cell (Figs. 1f, 3s; Supplementary Movie 2), it was quickly surrounded by vesicles containing bright DQ-BSA signals (Fig. 1f, starting from 24s). Shortly thereafter, it acquired a luminal fluorescent signal, indicating that vesicle fusion had

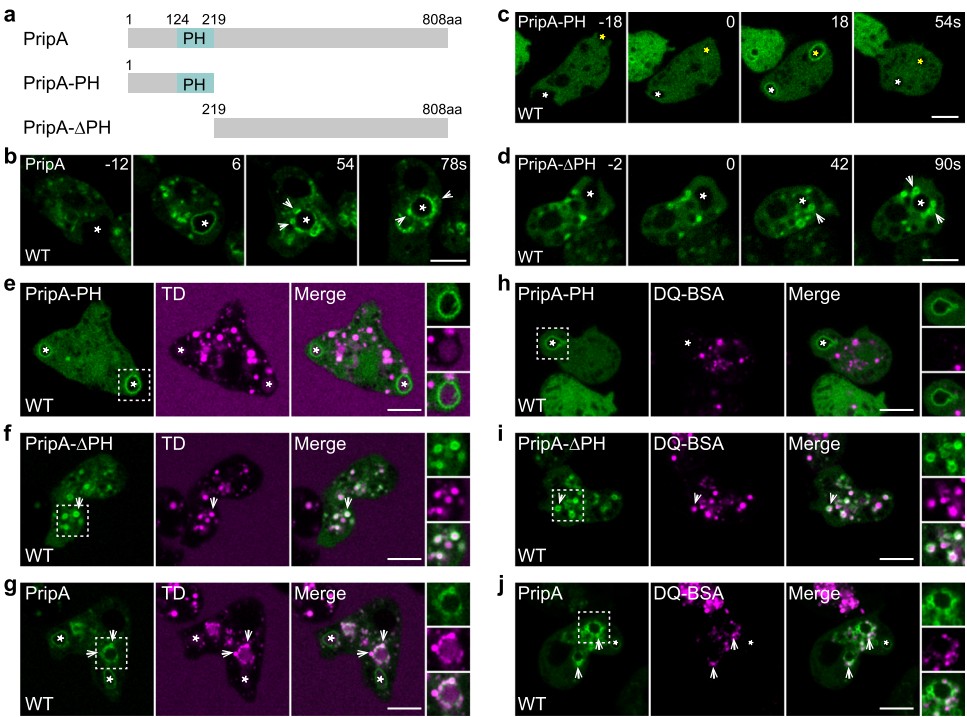

**Fig. 2 PripA localizes to macropinosomes. a** Schematic representation of PripA, PripA-PH, and PripA-ΔPH. **b–d** Time lapse images of PripA-GFP, PripA-PH-GFP, and PripA-ΔPH-GFP expressed in WT cells. The asterisks indicate newly formed macropinosomes. The arrows indicate smaller-sized vesicles labeled by PripA or PripA-ΔPH that surrounded newly formed macropinosomes. **e–g** TD uptake in WT cells expressing GFP-tagged PripA-PH, PripA-ΔPH, or PripA. The asterisks indicate newly formed macropinosomes containing weak TD signal. The arrows mark smaller macropinosomes labeled by PripA or PripA-ΔPH containing bright TD signal. **h–j** DQ-BSA degradation in WT cells expressing GFP-tagged PripA-PH, PripA-ΔPH, or PripA. The asterisks indicate newly formed macropinosomes containing minimal DQ-BSA signal. The arrows mark vesicles labeled by PripA or PripA-ΔPH containing bright DQ-BSA signal. Magnified images of the areas indicated by dashed boxes are shown on the right. Scale bar = 5 µm.

occurred to deliver cleaved DQ-BSA, and Rab5A was concurrently released from the macropinosomal membrane (Fig. 1f). A similar sequence of events was observed when using TD in the pulse-chase experiment (Supplementary Fig. S2c). Therefore, vesicle fusion is involved in macropinosome maturation during the conversion of Rab5 to Rab7.

**PripA is a PH domain-containing protein on macropinosomes**. Vesicle fusion may deliver regulatory components anchored on or enclosed in late macropinosomes to early macropinosomes, thereby promoting a rapid identity switch. To search for such regulatory elements, we screened PH domain-containing proteins that localize specifically on macropinosomes and identified PripA, an 808-amino-acid protein with a PH domain near the N-terminus (Fig. 2a). When expressed as a GFP fusion protein, a fraction of PripA was recruited onto macropinosomes approximately 5–10 s after membrane ruffles were closed (Fig. 2b, Supplementary Fig. S3a, Supplementary Movie 3). At a slightly later stage (approximately 25–30 s after ruffle closure), PripA-positive small vesicles started to gather around and eventually appeared to fuse with the newly formed macropinosomes as they moved further into the cell. Similar dynamic patterns were observed when PripA was expressed from a genome-integrated expression cassette (Supplementary Fig. S3b).

We generated truncation constructs to investigate the localization requirement of PripA. PripA-PH, which contains the N-terminal 219 amino acids including the PH domain (Fig. 2a), localized specifically on nascent macropinosomes, remained on the structures for a short period of time, and was then released into the cytoplasm (Fig. 2c, Supplementary Movie 3). Consistently, the PripA-PH-labeled vesicles contained weak TD and minimal DQ-BSA signals (Fig. 2e, h). In contrast, PripA-ΔPH, which contains the C-terminal portion of the protein (Fig. 2a), was absent from nascent macropinosomes but distributed on smaller cytoplasmic vesicles (Fig. 2d, Supplementary Movie 3). Some of these vesicles appeared to associate and fuse with newly formed macropinosomes, reminiscent of the Rab7-labeled late macropinosomes seen in Fig. 1e and Supplementary Fig. S2a. We confirmed that the PripA-ΔPH-labeled vesicles were late macropinosomes, as they contained bright luminal TD and DQ-BSA signals (Fig. 2f, i). In cells expressing full-length PripA, both types of vesicular structures could be visualized (Fig. 2g, j). Furthermore, PripA-positive vesicles with bright TD and DQ-BSA signals were often seen to position around newly entered macropinosomes (Fig. 2g, j). These experiments show that PripA is a macropinosome-associated protein with a distinct spatiotemporal pattern.

**PripA interacts with PI(3,4)P₂ and Rab7**. We investigated how the N- and C-terminal fragments of PripA control its localization on macropinosomes. Several lines of evidence indicate that PripA localizes to newly formed macropinosomes by interacting with PI(3,4)P₂ via the PH domain. Firstly, GFP-tagged PripA and PripA-PH, but not PripA-ΔPH, expressed in cell lysates bound specifically to PI(3,4)P₂ on lipid strips (Fig. 3a). Using lipid-coated agarose beads (Fig. 3b) and liposome flotation assays (Fig. 3c), we verified that the PH domain associated preferentially with PI(3,4)P₂. Secondly, using the PI(3,4)P₂ sensor TAPP1[41], we demonstrated that PripA-PH colocalized with TAPP1 on newly formed macropinosomes (Fig. 3d). The Pearson's correlation coefficient for PripA-PH-GFP and TAPP1-RFP was $0.71 \pm 0.09$ (mean ± s.e.m). Thirdly, in line with previous studies in

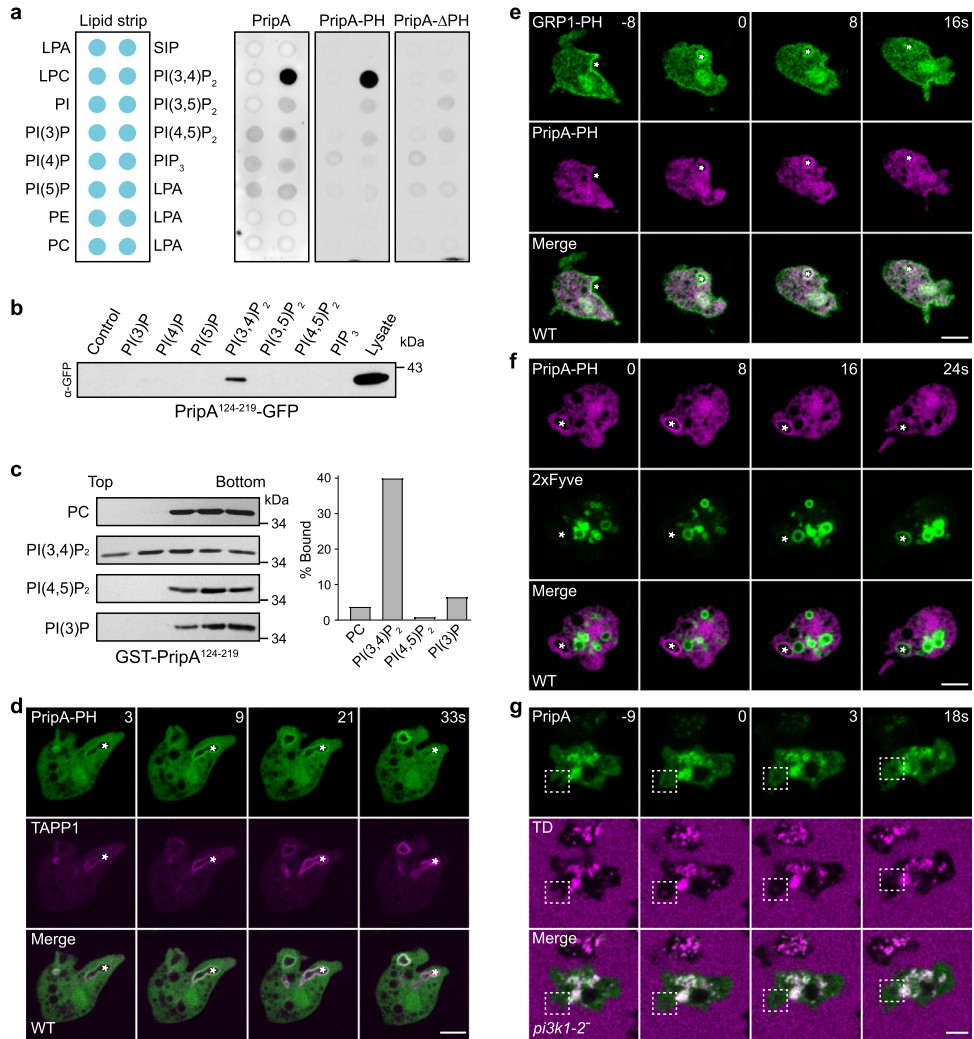

**Fig. 3 The PH domain of PripA interacts with PI(3,4)P₂.** **a** Lipid dot blot assay using cell lysates expressing GFP-tagged PripA, PripA-PH, or PripA-ΔPH. **b** Cells lysates expressing PripA[124-219]-GFP were incubated with the indicated lipid-coated agarose beads. Proteins eluted from the beads were probed with anti-GFP antibody. Lysate (0.8%) was loaded as input. **c** Left: liposome flotation assay with purified GST-PripA[124-219]. Proteins bound to liposomes were floated to the top of a sucrose gradient, from which five fractions were collected and analyzed by silver staining. Right: quantification of liposome binding. **d** Colocalization of PripA-PH-GFP and TAPP1-RFP on nascent macropinosomes. The Pearson's correlation coefficient was 0.71 ± 0.09 (mean ± SEM, $n = 8$ cells), where 1 = perfect, 0 = no correlation, and −1 = excluded. **e** Sequential accumulation of GRP1-PH-GFP and PripA-PH-RFP during macropinocytosis. **f** Sequential accumulation of PripA-PH-RFP and GFP-2 × Fyve during macropinocytosis. **g** Localization of PripA-GFP in *pi3k1-2⁻* cells incubated with TD. Nascent macropinosomes are marked by asterisks (**d–f**) or dashed boxes (**g**). Scale bar = 5 μm. Data in **a–c** was from one representative experiment out of two independent experiments. Source data for **a–c** are provided in this paper.

mammalian cells showing the sequential accumulation of PIP₃, PI(3,4)P₂, and PI3P during macropinosome formation and maturation[42,43], we found that PripA-PH was recruited to nascent macropinosomes shortly after the PIP₃ sensor GRP1-PH[44] but prior to the PI3P sensor 2 × Fyve[45] (Fig. 3e, f). A similar sequence was observed when imaging the membrane association kinetics of TAPP1 together with sensors for PIP₃ and PI3P (Supplementary Fig. S3c, d)[38]. Furthermore, in cells lacking the two major PI3-kinases responsible for producing PIP₃, which is quickly converted to PI(3,4)P₂ during macropinocytosis[38,42,43], the localization of PripA and TAPP1 on newly formed macropinosomes was specifically impaired (Fig. 3g, Supplementary Fig. S3e).

The size and distribution of PripA-ΔPH-positive vesicles resembled that of Rab7-marked macropinosomes. Therefore, we examined whether the localization of PripA-ΔPH depends on interaction with Rab GTPases, particularly Rab7. Using PripA or PripA-ΔPH as the prey in yeast two-hybrid (Y2H) assays, we screened 26 Rab GTPases with potential functions in endocytosis in their wild-type (WT) or constitutively active (CA) forms (Supplementary Table S1). PripA and PripA-ΔPH interacted selectively with the WT and CA forms of Rab7A and Rab7B (Fig. 4a, Supplementary Fig. S4a). We also observed interactions with Rab32A and Rab32D (Supplementary Fig. S4a). Imaging PripA-ΔPH and the four Rab proteins revealed that PripA-ΔPH exhibited substantial co-localization with Rab7A (Fig. 4b) but limited co-localization with Rab7B, Rab32A, and Rab32D (Supplementary Fig. S4b). The Pearson's correlation coefficient for PripA-PH-RFP and GFP-Rab7A was 0.86 ± 0.06 (mean ± s.e.m.). In addition, PripA and Rab7A, but not Rab5A, localized on enlarged macropinosomes resulting from the deletion of phosphoinositide 5-kinase *PIKfyve* (Supplementary Fig. S4c), which catalyzes the generation of PI(3,5)P₂[46]. Glutathione-S-transferase (GST) pull-down assays were performed to confirm the interaction between PripA-ΔPH and Rab7A. GFP-tagged PripA and PripA-ΔPH from cell lysates bound specifically to the

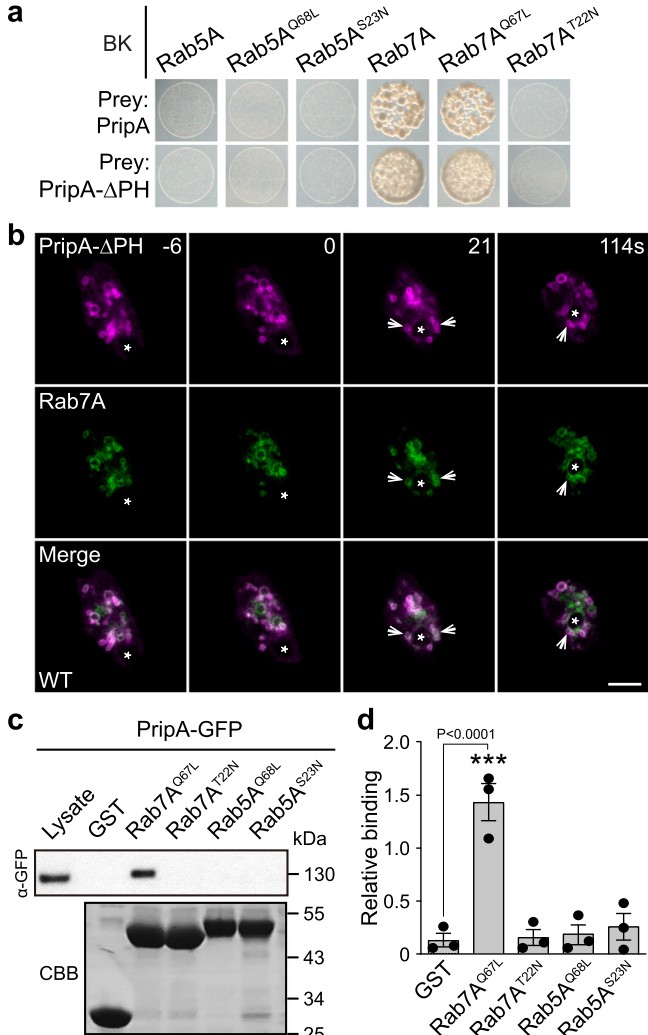

**Fig. 4 PripA interacts with the active form of Rab7. a** Yeast two-hybrid assay showing that PripA and PripA-ΔPH interacted specifically with WT and the constitutively active (CA) form of Rab7A (Rab7A$^{Q67L}$), but not the dominant negative (DN) form of Rab7A (Rab7A$^{T22N}$), Rab5A, the CA form of Rab5A (Rab5A$^{Q68L}$), or the DN form of Rab5A (Rab5A$^{S23N}$). **b** Colocalization of PripA-ΔPH-RFP and GFP-Rab7A on late macropinosomes. The Pearson's correlation coefficient was 0.86 ± 0.06 (mean ± SEM, $n = 8$ cells). **c** Top: Western blot from pull-down of GST control beads or GST-fused CA or DN forms of Rab5A and Rab7A beads with cell lysate expressing PripA-GFP. Bottom: the protein-transferred membrane was stained with coomassie brilliant blue (CBB) to show purified GST fusion proteins. **d** Densitometry showing relative binding of PripA-GFP to different GST fusion proteins (means ± SEM). Data were from 3 independent experiments. Significance was determined by one-way ANOVA with Dunnett posttest. Scale bar = 5 μm. Source data for **c**, **d** are provided in this paper.

CA form of Rab7A (Rab7A$^{Q67L}$) but not the dominant negative (DN) form of Rab7A (Rab7A$^{T22N}$) or either form of Rab5A (Rab5A$^{Q68L}$ and Rab5A$^{S23N}$) (Fig. 4c, d; Supplementary Fig. S4d). Therefore, PI(3,4)P$_2$ and Rab7 mediate the recruitment of PripA to macropinosomes.

**Dynamic localization of PripA during Rab5-to-Rab7 conversion.** The distinct localization of PripA and its ability to interact with both PI(3,4)P$_2$ and Rab7 suggest that PripA may link early and late macropinosomes and participate in the macropinosome maturation process. We imaged the sequential accumulation of PripA, Rab5, Rab7, and phosphoinositide signals in various combinations. In agreement with the early accumulation of PI(3,4)P$_2$ and Rab5A (Supplementary Fig. S5a), as well as the interaction between PripA-PH and PI(3,4)P$_2$ (Fig. 3a–d), we found that PripA-PH co-localized with Rab5A on macropinosomes (Fig. 5a). As newly formed Rab5A- or TAPP1-decorated macropinosomes dissociated from the plasma membrane, small PripA-ΔPH-positive vesicles arranged around and fused with them (Fig. 5b, c). This was much like the process observed during the replacement of Rab5A or TAPP1 by Rab7A (Fig. 1e, Supplementary Fig. S5b) and was consistent with the colocalization and interaction between PripA-ΔPH and Rab7A (Fig. 4). Imaging Rab5A or Rab7A together with PripA corroborated biphasic recruitment of PripA. PripA co-localized with Rab5A on newly formed macropinosomes. At a slightly later stage (approximately 20–30 s following cup closure), PripA- and Rab7A-positive small vesicles started to arrange around newly formed macropinosomes and fuse with them, which coincided with the loss of Rab5A and gain of Rab7A (Fig. 5d, e; Supplementary Movie 4, 5).

**PripA forms a complex with a putative Rab GAP protein, TbcrA.** To further elucidate the function of PripA in the context of the Rab5-to-Rab7 conversion, we attempted to identify its interactors by immunoprecipitating PripA-GFP from cell lysates, followed by mass spectrometry analysis of bound proteins. Among the candidate binding partners that associated specifically with PripA-GFP pull-down, a putative Rab GAP protein of 1194 amino acids caught our attention (Fig. 6a, b). We referred to this protein as TbcrA because it contains a TBC (Tre-2, Bub2, and Cdc16)/RabGAP domain at the C-terminus and a series of RCC1 domains at the N-terminus (Fig. 6a). Interestingly, using HMMER, a program designed to detect remote homologs, we found that the N-terminal and central regions of PripA contain stretches of sequences similar to human TBC1D2 (Supplementary Fig. S6, a–c), a Rab GAP localized on endosomes[47,48], though PripA does not contain a TBC domain at the C-terminus.

We verified the interaction between PripA and TbcrA in co-immunoprecipitation experiments. When co-expressed in cells, GFP-TbcrA could precipitate PripA-RFP from cell lysates (Fig. 6c). In addition to GFP-TbcrA, we generated two truncation constructs expressing either an N-terminal fragment of TbcrA containing the RCC1 domains (TbcrA-RCC1, 1-729 aa) or a C-terminal fragment containing the TBC domain (TbcrA-TBC, 722-1194 aa) (Fig. 6a). GFP-TbcrA-RCC1, but not GFP-TbcrA-TBC, specifically pulled down PripA-RFP (Fig. 6c). In addition, GFP-TbcrA-RCC1 was able to precipitate PripA-ΔPH-RFP, but not PripA-PH-RFP, from cell lysates (Fig. 6c). Therefore, the N-terminal region of TbcrA and the C-terminal region of PripA mediate the formation of a protein complex.

We observed the localization of TbcrA to examine its involvement in macropinocytosis. When expressed in WT cells, GFP-TbcrA localized to nascent macropinosomes and cytoplasmic vesicles similar to PripA (Supplementary Fig. S6d, e). In contrast, GFP-TbcrA-RCC1 mainly localized in the nucleus and GFP-TbcrA-TBC appeared to be cytosolic (Supplementary Fig. S6d). Interestingly, when co-expressed with PripA-RFP, GFP-TbcrA and GFP-TbcrA-RCC1, but not GFP-TbcrA-TBC, were strongly recruited to macropinocytic vesicles and co-localized with PripA (Fig. 6d, Supplementary Movie 6). The Pearson's correlation coefficient was 0.94 ± 0.01 (mean ± s.e.m.) for PripA and TbcrA, 0.91 ± 0.03 for PripA and TbcrA-RCC1, and 0.24 ± 0.03 for PripA and TbcrA-TBC. PripA-ΔPH-RFP was also able to recruit GFP-TbcrA-RCC1 (Fig. 6d). In contrast, expression of PripA-PH-RFP or VacA-RFP, a flotillin-like protein

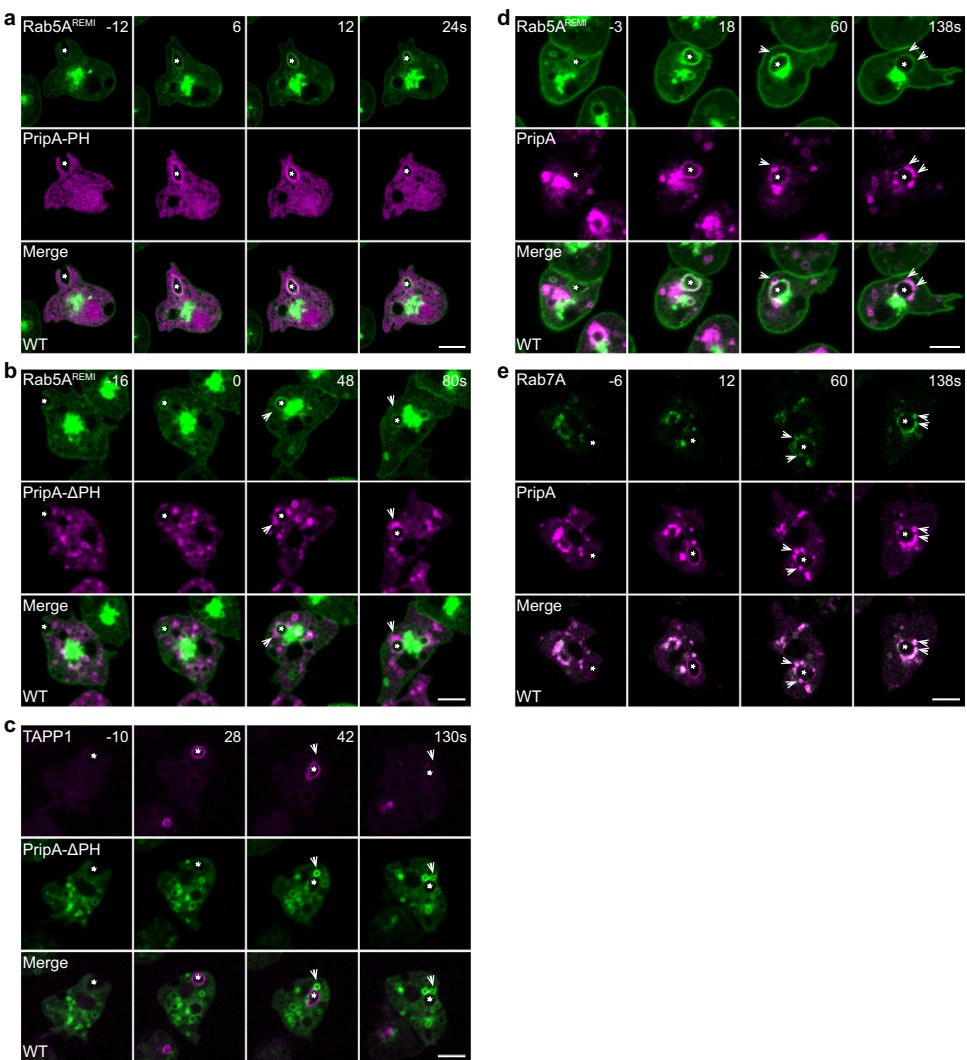

**Fig. 5 Time-lapse images of protein localization during macropinocytosis. a** Colocalization of GFP-Rab5A[REMI] and PripA-PH-RFP on nascent macropinosomes. **b** Localization of GFP-Rab5A[REMI] and PripA-ΔPH-RFP. **c** Localization of TAPP1-RFP and PripA-ΔPH-GFP. **d** Localization of GFP-Rab5A[REMI] and PripA-RFP. **e** Localization of GFP-Rab7A and PripA-RFP. In all panels, the asterisks mark newly formed macropinosomes and the arrows mark vesicles that gather around the newly formed macropinosomes. Scale bar = 5 μm.

found on postlysosomes and the plasma membrane, did not alter the localization of GFP-TbcrA-RCC1 (Fig. 6d). The Pearson's correlation coefficient value was 0.94 ± 0.03 for PripA-ΔPH and TbcrA-RCC1, 0.34 ± 0.12 for PripA-PH and TbcrA-RCC1, and 0.09 ± 0.04 for VacA and TbcrA-RCC1. These results correlate well with the co-immunoprecipitation experiments and suggest that PripA may complex with TbcrA to regulate macropinosome maturation.

**The PripA-TbcrA complex facilitates Rab5 inactivation**. Rab GAPs with TBC domain have been shown to facilitate GTP hydrolysis by supplying two catalytic residues[49]. We found that TbcrA contains these residues in two conserved motifs (Fig. 7a)[24,27], suggesting that it may function as an authentic Rab GAP. Using the aforementioned Y2H assay and TbcrA-TBC as the prey, we examined which Rab proteins may be targeted by TbcrA. Intriguingly, among the 26 Rab GTPases tested, TbcrA-TBC interacted specifically with Rab5A and Rab5B (Fig. 7b, Supplementary Fig. S6f). Consistent with studies showing that Rab proteins stabilized in an active conformation trap interactions with GAPs[24,27], we found that the interaction between TbcrA-TBC and Rab5A occurred preferentially with the CA form

(Rab5A[Q68L]) compared to the DN form (Rab5A[S23N]) (Fig. 7b). GST pull-down assays confirmed the interaction. GFP-TbcrA in cell lysates bound more robustly to the CA form than the DN form of Rab5A or either form of Rab7A (Fig. 7c, d). Interactions between TbcrA-TBC and RabJ were also detected by Y2H, but RabJ was not found on macropinocytic vesicles (Supplementary Fig. S6f, g). We performed Y2H assay on an additional 23 TBC domain-containing proteins encoded in the *Dictyostelium* genome to evaluate the binding to Rab5A (Supplementary Table S3) but did not detect any binding, verifying the interaction specificity between TbcrA and Rab5.

Using purified components and an optical GAP assay that continuously monitors the release of the inorganic phosphate resulting from GTP hydrolysis, we assessed the ability of TbcrA to promote GTP hydrolysis by Rab5A. Due to difficulties purifying full-length TbcrA and TbcrA-PripA complex, we examined the TBC domain instead. Purified TBC domain (820-1194 aa) accelerated GTP hydrolysis by Rab5A in a concentration-dependent manner (Fig. 7e). Furthermore, mutation of the conserved catalytic arginine and glutamine residues in the TBC domain to alanine reduced the GAP activity (Fig. 7e). In a global fit to a Michaelis–Menten model function, the catalytic efficiency

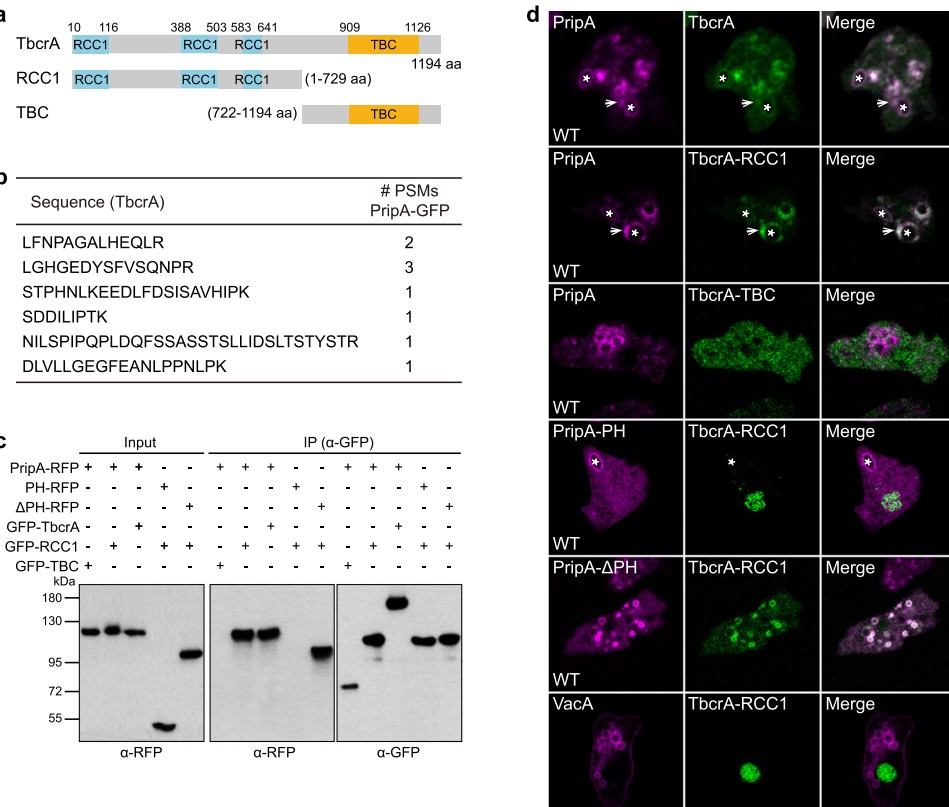

**Fig. 6 PripA forms a complex with TbcrA. a** Schematic representation of TbcrA, TbcrA-RCC1, and TbcrA-TBC. **b** Proteomic identification of TbcrA as a binding partner of PripA. The table shows the identified unique peptides of TbcrA from PripA-GFP, but not GFP control, pull-down. **c** Co-immunoprecipitation experiments showing that PripA and PripA-ΔPH specifically pulled down TbcrA and TbcrA-RCC1 from cell lysates. Data was from one representative experiment out of two independent experiments. **d** Co-expression of RFP-tagged PripA, PripA-PH, PripA-ΔPH, or VacA with GFP-tagged TbcrA, TbcrA-RCC1, or TbcrA-TBC. The Pearson's correlation coefficient was 0.94 ± 0.01 for PripA and TbcrA, 0.91 ± 0.03 for PripA and TbcrA-RCC1, 0.24 ± 0.03 for PripA and TbcrA-TBC, 0.33 ± 0.12 for PripA-PH and TbcrA-RCC1, 0.94 ± 0.03 for PripA-ΔPH and TbcrA-RCC1, and 0.09 ± 0.04 for VacA and TbcrA-RCC1. Data represents mean ± SEM (n = 10 cells). The asterisks mark newly formed macropinosomes and the arrows indicate surrounding vesicles. Scale bar = 5 μm. Source data for **b**, **c** are provided in this paper.

of TBC domain ($802.2 \pm 58.5\,M^{-1}s^{-1}$) was approximately twofold higher than that of mutated TBC domain ($357.2 \pm 40.3\,M^{-1}s^{-1}$). These experiments support the role of TbcrA as a GAP for Rab5 but, as the activity of the TBC domain in the in vitro GAP assay appears considerably weaker than that of previously characterized TBC domain proteins[49,50], we turned to cell experiments to further investigate the function of PripA-TbcrA complex in regulating Rab5 activity.

We measured the activation state of Rab5A in GFP-Rab5A[REMI]/WT cells or the same cells with deletion of *tbcrA* or *pripA* (Supplementary Fig. S7a–c). The Rab5-binding domain (1-209 aa) of human early-endosomal autoantigen 1 (EEA1), a well-known Rab5 effector that binds to active Rab5[30], was fused to GST, purified, and utilized to pull down GFP-Rab5A from cell lysates. Deletion of *tbcrA* led to a modest but significant increase in the amount of precipitated Rab5A, indicating that more Rab5A was in a GTP-bound state in *tbcrA⁻* cells (Fig. 7f, g). Consistently, when cells were transformed with EEA1[1-209]-RFP, an increased macropinosomal membrane association of EEA1 was detected in GFP-Rab5A[REMI]/*tbcrA⁻* cells compared to GFP-Rab5A[REMI]/WT cells, confirming that Rab5A inactivation was impaired in the mutant cells (Fig. 7h, i). Intriguingly, deletion of *pripA* caused almost identical defects in Rab5 inactivation as deletion of *tbcrA* (Fig. 7f–i). This finding is consistent with the results implying that PripA and TbcrA function in complex (Fig. 6) and the observation that the macropinosomal association of TbcrA was greatly impaired in

*pripA⁻* cells (Supplementary Fig. S6e). Deletion of *tbcrA* or *pripA* did not affect the size of EEA1-marked macropinosomes (Fig. 7j). Taken together, these results indicate that the PripA-TbcrA complex likely promotes Rab5 turnover during macropinocytosis by functioning as a GAP.

**The PripA-TbcrA complex promotes macropinosome maturation.** The above findings support a role of PripA-TbcrA complex in regulating macropinosome maturation by linking early and late macropinocytic compartments and promoting Rab conversion. These activities may help deliver components enclosed within late macropinosomes, such as hydrolytic enzymes, directly to early macropinosomes, thereby promoting cargo processing. We sought further evidence of defects in macropinosome maturation in the absence of the complex. Disruption of *pripA* or *tbcrA* or both genes resulted in reduced cell growth in liquid medium (Supplementary Fig. S7d), a process that relies on macropinocytic uptake of extracellular nutrients[32,51]. However, this mild phenotype also indicates that macropinosome maturation is not blocked in the mutant cells.

We examined the maturation process more carefully by following cargo processing. DQ-BSA degradation, which requires the macropinocytic compartments containing it to transit to the Rab7 stage and become hydrolytically active, was delayed in *pripA⁻* and *tbcrA⁻* cells (Fig. 8a, b). After a 30-min incubation

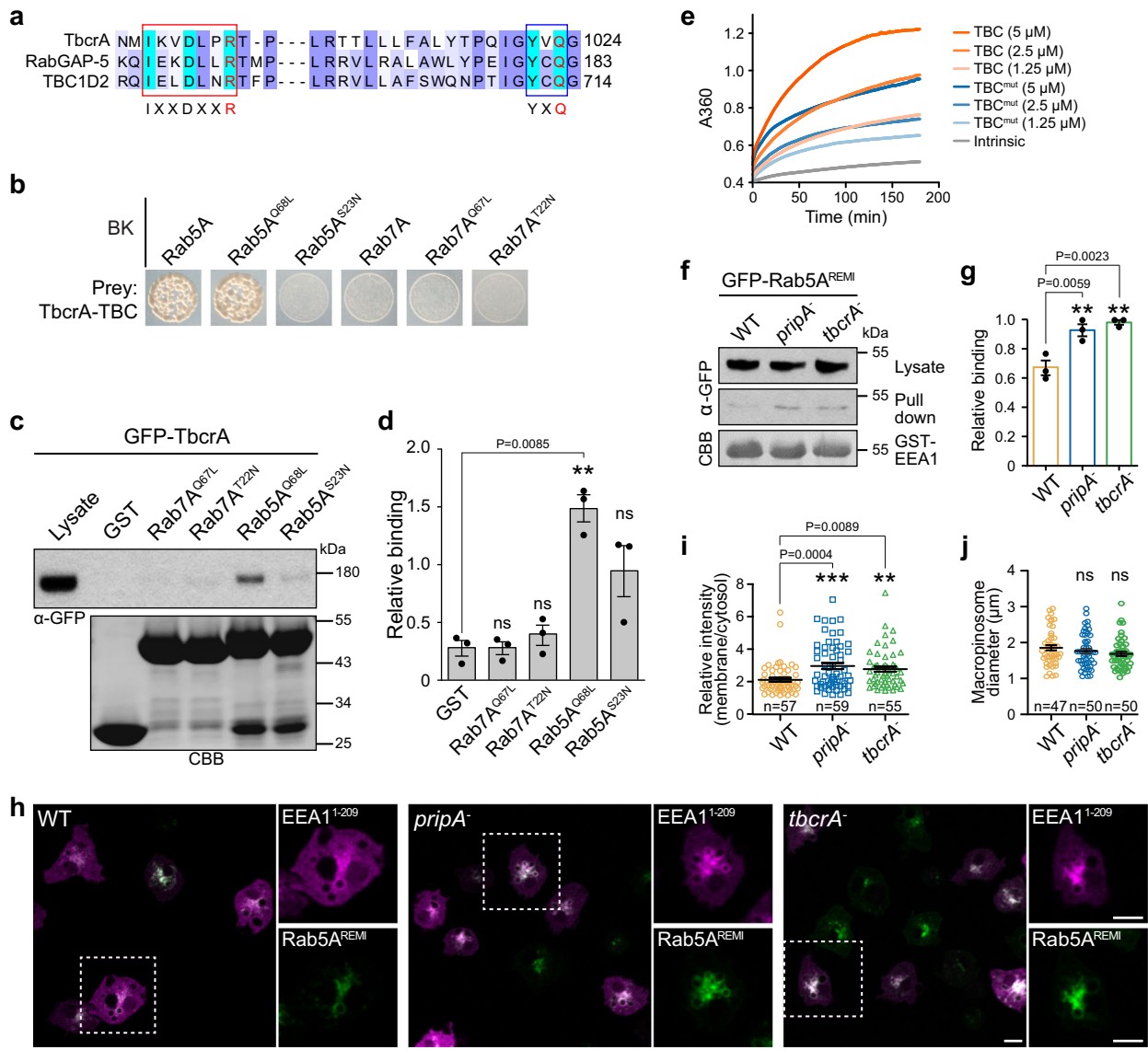

**Fig. 7 The PripA-TbcrA complex promotes Rab5 inactivation. a** Sequence alignment of TbcrA, human TBC1D2, and human RabGAP-5, spanning the region with conserved motifs important for GAP activity. The conserved arginine and glutamine residues are highlighted in red. **b** Yeast two-hybrid assay showing specific interaction between TbcrA-TBC and the WT or CA form of Rab5A. **c** Top: Western blot from pull down of GST or GST-fused CA or DN forms of Rab5A and Rab7A beads with lysates expressing GFP-TbcrA. Bottom: the protein-transferred membrane was stained with CBB to show purified GST fusions. **d** Densitometry showing relative binding of GFP-TbcrA to different GST fusions (means ± SEM from three independent experiments). **e** Time courses of GTP hydrolysis by Rab5A in the absence or presence of TBC domain or TBC domain-containing mutations in the conserved arginine and glutamine residues (TBC$^{mut}$). Graph shows one representative experiment. The catalytic efficiency was $802.2 \pm 58.5 \, M^{-1}s^{-1}$ for TBC domain and $357.2 \pm 40.3 \, M^{-1}s^{-1}$ for TBC$^{mut}$ (mean ± SEM from three independent experiments). **f** GST-EEA1$^{1-209}$ pull-down assays in GFP-Rab5A$^{REMI}$/WT cells or the same cells with deletion of *pripA* or *tbcrA*. The pull-down samples and lysates were probed with anti-GFP antibody. The protein-transferred membrane was stained with CBB to show purified GST-EEA1$^{1-209}$. **g** Quantification of the relative binding of GFP-Rab5A to GST-EEA1$^{1-209}$ beads in different cell lines (means ± SEM from three independent experiments). **h** Localization of EEA1$^{1-209}$-RFP in GFP-Rab5A$^{REMI}$/WT cells or the same cells with deletion of *pripA* or *tbcrA*. Magnified images of the areas indicated by dashed boxes are shown on the right. **i** Quantification of the fluorescent intensity of EEA1$^{1-209}$-RFP on the macropinosomal membrane relative to that in the cytosol. **j** Quantification of the maximum diameters of EEA1$^{1-209}$-RFP-marked macropinosomes. The scatter plots show data points with means and SEM. Significance was determined by one-way ANOVA with Dunnett posttest in all graphs. Scale bar = 5 μm. Source data for **c**–**g**, **i**, **j** are provided in this paper.

with DQ-BSA, many bright fluorescent spots were visible in WT cells, whereas the number and intensity of the spots were significantly reduced in knockout cells (Fig. 8a, b). A similar defect was observed by flow cytometry (Supplementary Fig. S8a). In contrast, TD uptake was not affected (Fig. 8c, d). These observations indicate that macropinosome maturation, but not formation, was specifically impaired by deletion of *pripA* or *tbcrA*. Notably, the size of TD- or DQ-BSA-containing macropinosomes

formed in the mutant cells was comparable to the size in WT cells (Supplementary Fig. S8b, c). Thus, the decreased DQ-BSA signal seen in the knockout cells was unlikely to be caused by macropinosome swelling, a consequence that may be expected of delayed Rab5 inactivation. In addition, defects in DQ-BSA processing were comparable in the double- and single-deletion mutants (Supplementary Fig. S8d, e), implying that PripA and TbcrA operate within the same pathway.

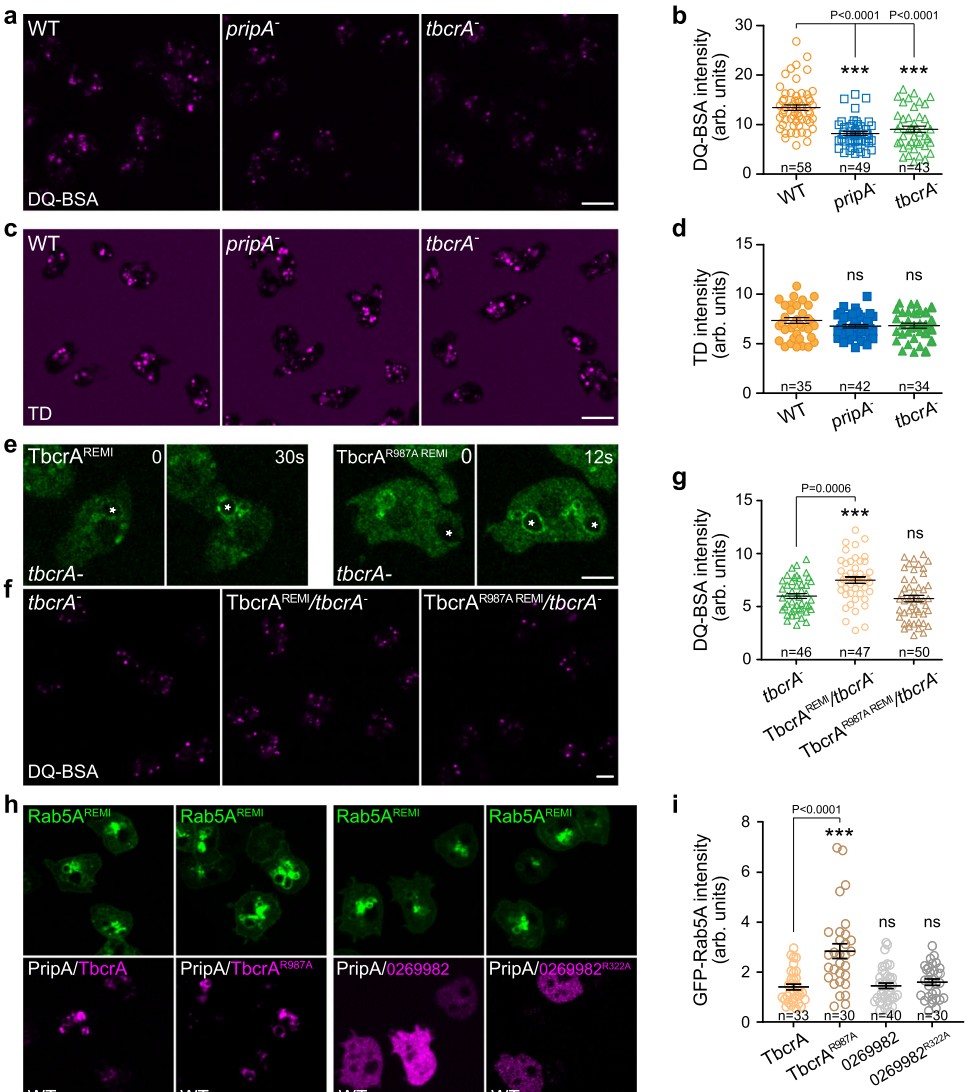

**Fig. 8 The PripA-TbcrA complex promotes macropinosome maturation. a** DQ-BSA degradation in WT, *pripA⁻*, and *tbcrA⁻* cells. Images were acquired after a 30-min incubation. **b** Quantification of DQ-BSA degradation. **c** TD uptake in WT, *pripA⁻*, and *tbcrA⁻* cells. Images were acquired after a 30-min incubation. **d** Quantification of TD uptake. **e** Localization of GFP-TbcrA and GFP-TbcrA$^{R987A}$ expressed from a stable copy integrated into the genome of *tbcrA⁻*. **f** DQ-BSA degradation in *tbcrA⁻*, GFP-TbcrA$^{REMI}$/*tbcrA⁻*, and GFP-TbcrA$^{R987A\ REMI}$/*tbcrA⁻* cells. Images were acquired after a 20-min incubation. **g** Quantification of DQ-BSA degradation. **h** GFP-Rab5A$^{REMI}$/WT cells were transformed with vectors expressing PripA together with RFP-TbcrA, RFP-TbcrA$^{R987A}$, RFP-DDB_G0269982, or RFP-DDB_G0269982$^{R322A}$. **i** Quantification of GFP-Rab5A signals on macropinosomes in the indicated cells lines. The scatter plots show data points with means and SEM. Significance was determined by one-way ANOVA with Dunnett posttest in all graphs. Scale bar = 5 μm. Source data for **b**, **d**, **g**, **i** are provided in this paper.

We examined whether the GAP activity of TbcrA is required for rescuing the DQ-BSA degradation defect observed in *tbcrA⁻* cells. To this end, we expressed GFP-tagged TbcrA or TbcrA containing a mutation in the catalytic arginine reside (TbcrA$^{R987A}$) in *tbcrA⁻* cells. The expression cassette was integrated into the genome as a single copy to ensure uniform expression in the cell population (Fig. 8e, Supplementary Fig. S8f). Expression of GFP-TbcrA, but not GFP-TbcrA$^{R987A}$, restored DQ-BSA processing activity (Fig. 8f, g), even though GFP-TbcrA$^{R987A}$ was able to localize to macropinosomes as its WT counterpart (Fig. 8e, Supplementary Fig. S8f). Furthermore, we found that co-expression of TbcrA$^{R987A}$, but not TbcrA, with PripA in GFP-Rab5A$^{REMI}$/WT cells led to the accumulation of enlarged macropinosomes labeled with GFP-Rab5A (Fig. 8h, i; Supplementary Fig. S8g), indicating a dominant effect of

TbcrA$^{R987A}$. A putative Rab GAP protein (DDB_G0269982) that localized in the cytoplasm was included as a control. Collectively, these results indicate that PripA and TbcrA function to modulate vesicle maturation along the macropinocytic pathway.

**The PripA-TbcrA complex and phagosome maturation**. Studies in *Dictyostelium* upon *Mycobacterium marinum* infection have implied that phagosomes acquire Rab5 and Rab7 in a sequential manner[52,53]. Interestingly, we found that the transition of Rab5 to Rab7 on phagosomal membranes involves the association and fusion of Rab5-marked phagosomes with smaller Rab7-marked vesicles (Supplementary Fig. S9a, Supplementary Movie 7), similar to the sequence of events occurring during macropinocytosis. PripA and TbcrA were also recruited to phagosomes (Supplementary Fig. S9b, c; Supplementary Movie 8), and

deletion of *pripA* impaired the phagosomal association of TbcRA (Supplementary Fig. S9c, d). Furthermore, although *pripA* or *tbcrA* deletion did not affect bacterial phagocytosis-dependent cell growth, the mutant cells exhibited compromised Rab5 inactivation and delayed phagocytic cargo degradation (Supplementary Figs. S7e and S10), indicating that the PripA-TbcrA complex also regulates vesicle maturation along the phagocytic pathway.

## Discussion

In this study, we uncovered a protein complex composed of PripA and TbcrA that regulates macropinosome maturation during the Rab5-to-Rab7 transition stage. We show that transition of Rab5 to Rab7 is mediated, at least in part, by fusion of Rab5-marked early macropinosomes with Rab7-marked late macropinosomes. During this process, PripA likely links the two membrane compartments by interacting with PI(3,4)P₂ and Rab7. In addition, PripA recruits TbcrA, which in turn functions as a GAP, to promote Rab5 inactivation. This working mechanism is in line with the Rab GAP cascade model, which argues that the activation of a downstream Rab serves to recruit GAP for the upstream Rab, thereby limiting the extent of overlap between adjacent Rab compartments[54]. In our pathway, activated Rab7 recruits TbcrA through PripA to control the duration of Rab5 activation during macropinocytosis. Thus, the PripA-TbcrA complex functions as the central component of a Rab GAP cascade to modulate Rab conversion and macropinosome maturation in *Dictyostelium* (Fig. 9). Intriguingly, a previous study on yeast endocytosis suggested a similar model wherein inactivation of the yeast Rab5 requires not only an associated GAP but activation of Rab7 and endosome-vacuole fusion[25,26]. Together with other examples[55–57], these studies indicate that Rab GAP cascade may be a general mechanism for generating a programmed switch in the Rab association on membrane compartments as they move along endocytic or exocytic pathways.

The conversion from Rab5 to Rab7 appears to be important for macropinosome maturation and cargo processing. We found that disruption of either *pripA* or *tbcrA* delayed Rab5 inactivation and DQ-BSA degradation but did not affect macropinosome formation measured by the uptake of 70 kDa dextran. Consistently,

expression of the CA form of Rab5A in WT cells led to defects in DQ-BSA degradation but not TD internalization (Supplementary Fig. S11). In contrast, expression of the DN form of Rab5A markedly reduced TD uptake (Supplementary Fig. S11). This data agrees with a recently published study showing that Rab5 activation plays a critical role in promoting macropinosome sealing and scission[58]. We also found that expression of the DN form of Rab7A inhibited TD uptake (Supplementary Fig. S11), which is consistent with earlier reports in *Dictyostelium* and other systems[59,60], but the molecular mechanism underlying this defect is not fully understood. These findings highlight the importance of precisely controlling the activation and turnover of Rab5 and Rab7 during endocytic trafficking.

The detailed mechanism of how activities of the PripA-TbcrA complex are controlled during macropinocytosis requires further investigation. There appears to be biphasic recruitment of PripA and TbcrA. The initial phase begins immediately after ruffle closure, leading to accumulation of the complex on nascent macropinosomes. This event likely depends on interactions between PripA and PI(3,4)P₂. In the absence of the PH domain that binds to PI(3,4)P₂, the early recruitment of PripA is specifically impaired. The second phase occurs following tethering and fusion of the newly formed macropinosomes with late macropinosomes positive for Rab7, PripA, and TbcrA. Notably, the macropinosomal localization of TbcrA was not abolished in *pripA⁻* cells, suggesting that additional signals may contribute to its recruitment. Furthermore, macropinosomes marked by PripA-ΔPH can also associate with nascent macropinosomes. Thus, although PripA possesses the ability to link early and late macropinosomes, additional factors likely exist to reinforce such an association (Fig. 9). We do not yet understand why PripA and TbcrA translocate to macropinosomes in two stages. It is somewhat counterintuitive to observe them early on, considering that Rab5 activity is important for many early events[15–17,61]. It is possible that the PripA-TbcrA complex is required to sharpen the effect of GEF-mediated Rab5 activation even at the early stage. Alternatively, the complex may not become active until further recruitment following membrane fusion. In this scenario, tethering and fusion with late macropinosomes may permit simultaneous interaction among PripA, active Rab7, and TbcrA, which

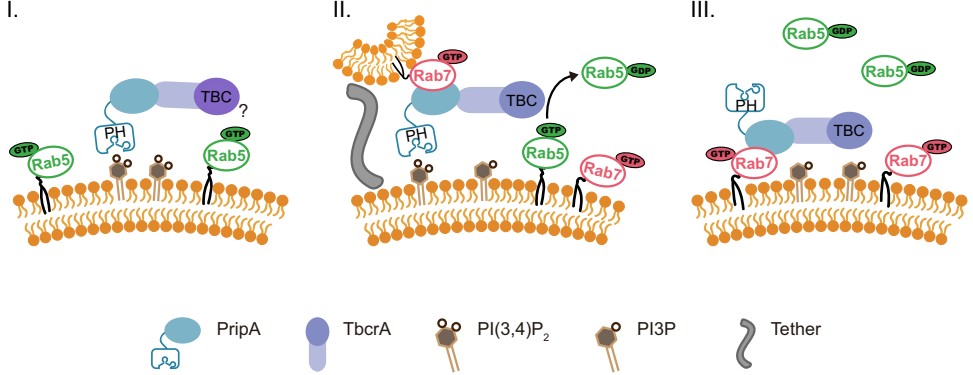

**Fig. 9 Schematic of Rab5-to-Rab7 conversion modulated by the PripA-TbcrA complex.** The PripA-TbcrA complex acts as the central component of a Rab GAP cascade regulating macropinosome maturation in *Dictyostelium*. The initial stage (I) of macropinosome maturation occurs shortly after ruffle closure, in which PI(3,4)P₂ on newly formed Rab5-enriched macropinosomes recruits PripA and TbcrA. The PripA-TbcrA complex may be required to sharpen Rab5 activation at this early stage, or alternatively, it may not yet become fully active (indicted by the question mark). In the second stage (II), tethering and fusion of Rab7-positive late macropinosomes with early macropinosomes mediate additional PripA-TbcrA recruitment. The ability of PripA to interact with both PI(3,4)P₂ and Rab7 may link the two membrane compartments, and other tethering factors yet to be identified likely reinforce such association. At the same time, interactions among active Rab7, PripA, and TbcrA may stimulate the GAP activity of TbcrA to promote Rab5-GTP hydrolysis. In this way, Rab5 is turned off when the PripA-TbcrA complex responds to Rab7-GTP, which ensures consecutive function of Rab5 and Rab7. In the final stage (III), early macropinosomes mature into late macropinosomes marked by Rab7. The macropinosome maturation process is accompanied by the degradation of PI(3,4)P₂ to PI3P.

may somehow stimulate the GAP activity of TbcrA. In this way, Rab5 is not turned off until the PripA-TbcrA complex responds to Rab7-GTP. To prove this model will require future studies to develop a better Rab5 activity sensor (the human EEA1 protein generates rather diffuse signal in WT cells) in order to track the level of Rab5 activation in real time. It will also require purification of full-length PripA and TbcrA and comparison of the GAP activity of the complex in the presence and absence of activated Rab7.

Additional questions remained to be answered. First, whether membrane fusion and a similar Rab GAP cascade regulate macropinosome maturation in other systems requires further investigation. Interestingly, in HT1080 human fibrosarcoma cells, which have a constitutively high level of macropinocytosis similar to *Dictyostelium*, the transition from RAB5 to RAB7 appears to follow the association and fusion of RAB5-marked macropinosomes with RAB7-marked vesicles (Supplementary Fig. S12, Supplementary Movie 9), reminiscent of what occurs in *Dictyostelium* cells. Second, macropinocytosis and phagocytosis have been shown to share similar, though not identical, regulatory mechanisms[62,63]. PripA and TbcrA also appear to regulate phagosome maturation. The similarities and differences in how they act in different endocytic pathways requires further elucidation. Third, because deletion of the PripA-TbcrA complex does not completely block macropinosome maturation or phagosome maturation, which may be due to yet unknown redundancy in the pathways, future studies are needed to identify additional molecular machineries regulating these processes.

## Methods

**Dictyostelium cell culture**. WT cells were derived from the Ax2 axenic strain provided by the Robert Kay laboratory (MRC Laboratory of Molecular Biology, London, UK). All gene deletion cell lines were generated in this Ax2 background. WT and gene deletion cells were cultured in HL5 medium (Formedium # HLF3) supplemented with antibiotics at 22 °C. Cells carrying expression constructs were maintained in HL5 containing G418 (10–20 μg/ml) or Hygromycin (50 μg/ml).

**Gene disruption, plasmid construction, and stable expression**. Plasmids and primers used in this study are listed in Supplementary Tables S1 and S2. To make knockout constructs for *pripA*, *tbcrA*, or *pikfyve* deletion, a Blasticidin S resistance (BSR) cassette was inserted into pBlueScript II SK+ to generate pBlueScript-BSR. 5′ and 3′ arms were PCR-amplified from genomic DNA with primers listed in Supplementary Table S2 and cloned upstream and downstream of the BSR cassette, respectively. The resulting disruption cassette was PCR-amplified or digested from the knockout construct and electroporated into Ax2. Gene disruption was confirmed by resistance to Blasticidin (10 μg/ml), PCR and Southern blot. To generate *pripA* and *tbcrA* double knockout cells, the BSR cassette was first removed from *pripA⁻* cells by transformation with a Cre recombinase expression plasmid, pDEX-NLS-Cre[64], and selection with 20 μg/ml G418. The *tbcrA* gene was then disrupted.

To generate constructs expressing GFP- or RFP-fusion proteins, DNA fragments encoding full-length or truncations of PripA, TbcrA, Rab GTPases, and reporter proteins were PCR-amplified using primers listed in Supplementary Table S2 and cloned into *Dictyostelium* expression vectors[65] containing a multiple cloning site. To generate EEA1¹⁻²⁰⁹-RFP, the sequence corresponding to human EEA1 (1-209 aa) was *Dictyostelium* codon optimized and cloned into pDM451. To generate constructs for co-expression of PripA and TbcrA or PripA and TbcrAᴿ⁹⁸⁷ᴬ, *pripA* was first cloned into pDM344, and the expression cassette was subsequently released by digestion with NgoMIV and cloned into pDM449. The *tbcrA* and mutated *tbcrA* genes were then cloned into the same vector. To generate construct expressing GFP-2 × FYVE, 2 × FYVE domain of *C. elegans* EEA1 was released from a *C. elegans* expression vector and cloned into pDM317.

To generate constructs for genome-integrated expression, the sequence between NgoMIV and HindIII in the extrachromosomal expression vectors was removed. The resulting fragments were gel purified, blunt using NEB Quick Blunting kit, and re-ligated. The constructs were then linearized with XhoI and electroporated into the appropriate cells together with XhoI. Clones were selected with G418. Stability of expression was examined by imaging after single cell expansion with bacteria for multiple generations. At least three independent clones were generated for each cell line, their phenotypes were compared, and results from one representative clone are shown.

To generate constructs expressing GST-fusion proteins, cDNA fragments encoding Rab5A, Rab7A, and PripA¹²⁴⁻²¹⁹ were PCR-amplified and cloned into pGEX-4T-1 or pGEX-6P-1. To generate GST-TBC⁸²⁰⁻¹¹⁹⁴ and mutated GST-

TBC⁸²⁰⁻¹¹⁹⁴, the sequence of TBC was codon optimized and then cloned into pGEX-6P-1. To generate constructs expressing His-MBP-Rab5A, cDNA fragment of Rab5A was PCR-amplified and cloned into pET-MBP-3C. To generate construct for expressing mCherry-RAB5A in HT1080 cells, mCherry and RAB5A were PCR-amplified from pmCherry-C1 and pEGFP-C2-RAB5A and cloned into pCDH-CMV.

**Imaging**. To image the localization of fluorescent proteins in *Dictyostelium* cells, 10⁵ cells were plated in 8-well coverslip chamber (Lab-Tek, NalgenNunc) and allowed to adhere. When necessary, cells were cultured in SIH or LoFlo medium (Formedium) to reduce autofluorescence. Images were taken on a Zeiss 880 inverted microscope or a Zeiss 980 Airyscan confocal microscope equipped with a Fastscan detector using a 63 ×/1.4 oil-immersion objective. For experiments in Figs. 1a–d, 2h–j, and Supplementary Fig. S1a, cells were incubated in LoFlo medium containing 500 μg/ml TRITC-dextran (TD) (Sigma #T1162) or 10 μg/ml DQ-BSA (Invitrogen #D-12050). For experiments in Figs. 1f and Supplementary Fig. S2c, cells were incubated in medium containing 25 μg/ml DQ-BSA or 500 μg/ml TD for 60 min (DQ-BSA and TD were removed before imaging). For experiments in Fig. 2e–g, cells were incubated in LoFlo medium containing 500 μg/ml TD and 2% unlabeled dextran (Sigma #D3759) for 60 min. For experiments in Fig. 8a, c, f, and Supplementary Fig. S8d, cells were incubated in SIH medium containing 10 μg/ml DQ-BSA or 500 μg/ml TD. For experiments in Supplementary Figs. S1e–i and S11, cells were incubated in HL5 medium containing 500 μg/ml TD or 10 μg/ml DQ-BSA. All image analyses were performed using ImageJ 1.45j. Time evolution of the fluorescent intensity during macropinosome or phagosome maturation was determined by recording the medial optical section. For each channel, the average intensity at the macropinosomal or phagosomal membrane was determined after background subtraction; the value was divided by the average fluorescence intensity in the cytosol to obtain the membrane-to-cytosol ratio. The size distribution of macropinosomes was analyzed with the analyze particles tool in ImageJ.

**Immunoprecipitation assay and mass spectrometry analysis**. To identify proteins that interact specifically with PripA, cells expressing the GFP control or PripA-GFP were lysed by ice-cold lysis buffer (10 mM NaPi pH 7.2, 100 mM NaCl, 0.5% Nonidet P-40, 10% glycerol, 1 mM NaF, 0.5 mM Na₃VO₄, 1× complete EDTA-free protease inhibitor (Roche) and 1 mM DTT) and incubated for 5 min on ice. Lysates were centrifuged at 22,000 × g for 5 min at 4 °C. The supernatants were incubated with anti-GFP affinity beads (Smart Lifesciences #SA070005) for 2 h at 4 °C. Beads were washed three times with lysis buffer. Samples were eluted with SDS loading buffer and subjected to SDS-PAGE. Protein bands were visualized by CBB staining. Each gel lane was divided into three slices and subjected to in-gel trypsin digestion and mass spectrometry analysis as described before[38]. The mass spectrometry data have been deposited to the ProteomeXchange Consortium via the PRIDE[66] partner repository with the dataset identifier PXD032072.

For co-immunoprecipitation experiments presented in Fig. 6c, cells expressing GFP- or RFP-tagged full length or truncated TbcrA or PripA were lysed by ice-cold lysis buffer (20 mM Hepes pH 7.2, 100 mM NaCl, 0.3% CHAPS, 1 mM NaF, 0.5 mM Na₃VO₄, 1 mM DTT, and 1× complete EDTA-free protease inhibitor) and incubated for 10 min on ice. Lysates were centrifuged at 22,000 × g for 5 min at 4 °C. The supernatants were incubated with anti-GFP affinity beads for 2 h at 4 °C. Beads were washed three times with lysis buffer. Samples were eluted with SDS loading buffer and subjected to SDS-PAGE. Western blotting was carried out as described before[67]. Anti-GFP antibody (WB, 1:5000) was purchased from Roche (#11814460001). Anti-DsRed antibody (WB, 1:1000), which was used to detect RFP-fusion proteins, was purchased from TaKaRa (#632496).

**Lipid-protein interaction assays**. Lipid dot blot assay and lipid-coated agarose bead pull-down assay were performed as described before[38]. For liposome flotation assay, POPC (850457), NBD-PE (810145), PI3P (850150), PI(3,4)P₂ (850153), and PI(4,5)P₂ (850155) were obtained from Avanti Polar Lipids. POPC, NBD-PE, and variable PIPs were mixed at molar ratio of 89: 1: 10. Mixed lipids were dried under a flow of nitrogen gas and in SpeedVac for 1–2 h. The lipid films were resuspended in Hepes-NaCl buffer (50 mM Hepes pH 7.2 and 150 mM NaCl) to a final concentration of 5 mM and subjected to freeze-thaw cycles 11 times. Unilamellar liposomes were generated via extrusion through a nanopore membrane with a pore size of 100 nm (Avanti Polar Lipids #610005); the process was repeated 11 times. The liposomes were mixed with purified proteins at molar ratio of 1000: 1 in a 50 μl reaction and incubated at 4 °C for 1 h with gentle agitation. 30 μl of the protein-liposome mixture was diluted with 100 μl 1.9 M sucrose, placed at the bottom of a centrifugation tube, and overlaid sequentially with 100 μl 1.25 M sucrose and 20 μl Hepes-NaCl buffer. The sucrose gradient samples were centrifuged at 174,000 × g for 1 h at 4 °C. Five fractions were collected from the top, mixed with SDS loading buffer, and subjected to SDS-PAGE followed by silver staining. Relative binding was calculated as the sum of band intensities of top two fractions divided by the sum of all five fractions.

**Purification of recombinant proteins**. Expression of GST or His-MBP fusion proteins was induced in *E.coli* strain BL21 (DE3) (Biomed #BC201) with 0.3 mM isopropyl β-D-1-thiogalactoside (IPTG) at 16 °C overnight. To purify GST fusion proteins used in pull-down assays, the bacteria pellet was resuspended in PBS

supplemented with protease inhibitor (Bimake #B14012) and sonicated, followed by a 30 min spin at $18,000 \times g$ to remove debris. The supernatant was incubated with glutathione sepharose beads (GE #17-0756-05) at $4\,°C$ for 2 h. The beads were washed with buffer (20 mM Tris-HCl pH 7.5 and 300 mM NaCl) and stored at $-80\,°C$ in buffer (20 mM Tris-HCl pH 7.5, 150 mM NaCl, 1 mM DTT, and 50% glycerol). For purification of GST-Rab5A and -Rab7A, all buffers were supplemented with 5 mM $MgCl_2$.

To purify GST-tagged PripA-PH and TbcrA-TBC, the bacteria pellet was resuspended in buffer (50 mM Hepes pH 7.2, 150 mM NaCl, protease inhibitor, 1 mM PMSF, and 5 mM βME) and then lysed by high-pressure cell breaker, followed by a spin to remove debris. The supernatant was incubated with glutathione sepharose beads at $4\,°C$ for 1 hr. The beads were washed with buffer (50 mM Hepes and 150 mM NaCl). Proteins were eluted with 20 mM Tris-HCl pH7.2 containing 10 mM reduced Glutathione, concentrated and buffer exchanged using centrifugal filter unit (Millipore #UFC803024), and finally stored at $-80\,°C$ in buffer containing 50 mM Hepes, 150 mM NaCl, and 1 mM DTT.

To purify His-MBP fusion proteins, the bacteria pellet was resuspended in buffer (20 mM Hepes pH 7.2, 150 mM NaCl, protease inhibitor, 1 mM PMSF, 10 mM Imidazole, and 5 mM βME) and then lysed by cell breaker, followed by a spin to remove debris. The supernatant was incubated nickel-NTA agarose (Sigma #H0537) at $4\,°C$ for 1 hr. The beads were washed with wash buffer (20 mM Hepes, 300 mM NaCl, 20 mM Imidazole, and 5 mM βME) and binding buffer (20 mM Hepes, 150 mM NaCl, 10 mM Imidazole and 5 mM βME). Proteins were eluted by elution buffer (20 mM Hepes, 150 mM NaCl, 300 mM Imidazole, and 5 mM βME). Desalting column (GE healthcare #17085101) was used to remove extra Imidazole, and eluted proteins were stored at $-80\,°C$ in buffer containing 20 mM Hepes, 150 mM NaCl, and 1 mM DTT.

**GST pull-down assay.** For pull-down assays presented in Figs. 4c, 7c, f, cells expressing GFP-fusion proteins were lysed at $4 \times 10^7$ cells/ml with ice-cold lysis buffer (20 mM Hepes pH 7.2, 100 mM NaCl, 0.3% CHAPS, 5 mM $MgCl_2$, 1 mM DTT and complete EDTA-free protease inhibitor) and incubated for 10 min on ice. Lysates were centrifuged at $22,000 \times g$ for 10 min. The supernatants were incubated with pre-washed glutathione sepharose beads containing 15 μg GST-Rab proteins at $4\,°C$ for 2 h. After the incubation, beads were washed four times with lysis buffer. Samples were eluted with SDS loading buffer and subjected to SDS-PAGE.

**GAP assay.** His-MBP-Rab5A was loaded with a 25-fold molar excess of GTP at $25\,°C$ for 30 min in 20 mM Hepes pH 7.2, 150 mM NaCl, 5 mM EDTA, and 1 mM DTT. Free nucleotide was removed by a desalting column (ThermoFisher #89882) pre-equilibrated with ice-cold buffer containing 20 mM Hepes, 150 mM NaCl, and 10 mM $MgCl_2$. GTP hydrolysis was measured by using the EnzChek Phosphate Assay Kit (Invitrogen #E6645). The GAP reaction mixture contains 20 mM Hepes, 150 mM NaCl, 0.2 mM 2-amino-6-mercapto-7-methylpurine ribonucleoside, 1 U/ml of purine nucleoside phosphorylase, 10 mM $MgCl_2$, 20 μM GTP-loaded Rab5A, and various concentrations of GAPs. The absorbance at 360 nm was recorded every 30 s using a microplate spectrometer (TECAN). Data were analyzed by fitting to the pseudo-first-order Michaelis–Menten model function $A(t) = (A_\infty - A_0)(1 - \exp(-k_{obs}t)) + A_0$ where $k_{obs} = k_{intr} + (k_{cat}/K_m)[GAP]$ as described before[49]. The catalytic efficiency ($k_{cat}/K_m$) and intrinsic rate for GTP hydrolysis ($k_{intr}$) were treated as global parameters.

**Yeast two-hybrid assay.** To analyze protein interactions in yeast, *S. cerevisiae* strain AH109 and Matchmaker GAL4 Two-Hybrid System 3 (Clontech Laboratories) was used. PripA, PripA-ΔPH, and TbcrA-TBC coding DNA sequences (CDSs) were PCR-amplified using primers listed in Supplementary Table S2 and cloned into pGADT7 vector in frame with CDS for Gal4 activation domain (AD). CDSs for WT or CA forms of Rab GTPases were PCR-amplified using primers listed in Supplementary Table S2 and cloned in frame with Gal4 DNA binding domain in pGBKT7 vector. Yeast cells were cotransfected with both bait and prey plasmids and interactions between tested proteins were analyzed according to the manufacturer's instructions.

**Flow cytometry assay.** For flow cytometry analysis in the Supplementary Fig. S8a, $2$–$3 \times 10^6$ cells seeded in 6-well plate were incubated in SIH medium supplemented with 10 μg/ml DQ-BSA for 20 min. After incubation, cells were washed with ice-cold KK2 buffer (6.5 mM $KH_2PO_4$, 3.8 mM $K_2HPO_4$, pH 6.2) containing 10 mM EDTA and resuspended in KK2 buffer containing 5 mM sodium azide. The total fluorescence intensity per cell was analyzed by Attune NxT flow cytometer (ThermoFisher) with gating strategies shown in Supplementary Fig. S13.

**Phagocytosis assay.** To image yeast phagocytosis, $10^5$ cells were plated in 8-well coverslip chamber and allowed to adhere. After cell adherence, old medium was replaced by fresh medium containing $10^6$ yeast particles, and images were taken every 14 s for 20 min. For bacteria-killing assay, expression of GST-GFP was induced in *E.coli* strain BL21 (DE3) with 0.3 mM IPTG at $16\,°C$ overnight. The bacteria pellet was resuspended in SIH medium to a concentration of $2 \times 10^8$ bacteria/ml. 10 μl of bacteria culture in 400 μl SIH was added into 8-well coverslip chamber before addition of $10^5$ cells. Images were acquired every 14 s for a total of 120 frames. To measure the doubling time of cell growth using bacteria as food, cells were grown in 6-well plate starting at $10^5$ cells for 20 h in bacterial suspensions diluted to $OD_{600} = 2$ in SorMC buffer [15 mM $KH_2PO_4$, 2 mM $Na_2HPO_4$, 50 μM $MgCl_2$, 50 μM $CaCl_2$ (pH 6.0)].

**Mammalian cell culture.** HT 1080 cells were cultured in MEM without Glutamine (ThermoFisher #11090081), supplemented with 10% FBS (Gibco #10091148), GlutaMAX (Gibco #35050061), MEM NEAA (Gibco #11140050), Sodium Pytuvate (Gibco #11360070) and 1% Penicillin-Streptomycin (Gibco #15140122). HEK293T cells were cultured in DMEM (HyClone #SH30022.01) containing 10% FBS and 1% Penicillin-Streptomycin. All cell lines were maintained at $37\,°C$ and under 5% $CO_2$.

To generate HT1080 cells stably expressing mCherry-RAB5A and mEmerald-RAB7A, lentiviral particles were prepared in HEK293T cells. In brief, cells plated in 6 well plates were transfected with pCDH-mCherry-RAB5A or pCDH-mEmerald-RAB7A (1 μg), together with packaging plasmid psPAX2 (750 ng) and envelop plasmid pDM2.G (250 ng), using Lipofectamine 3000 (ThermoFisher #L3000015). Supernatants containing viral particles were collected 48 h after transfection, filtered through 0.45 μm membrane (Millipore #SLHV033RB), and mixed with polybrene (YEASEN #40804ES86) for a final concentration of 10 μg/ml before infecting HT1080 cells. Stably transfected HT1080 cells were selected with 1 μg/ml of puromycin.

**Statistics and reproducibility.** Statistical analysis was performed using GraphPad Prism 9.0. Statistical significance was determined by one-way ANOVA with Dunnett post-test or two-tailed unpaired t test with Welch's correction. In all figures, *** indicates $p < 0.001$, **$p < 0.01$, and *$p < 0.05$. All microscopy experiments were repeated independently at least three times with representative images shown in Figs. 1a–f; 2b–j; 3d–g; 4b; 5; 6d; 7h; 8a, c, e, f, h and in Supplementary Figs. 1a–d; 2a, c; 3a–c, e; 4b–c; 5; 6d–e; 8d, f, g; 9a–c; 10a, d; 12. Experiment presented in Supplementary Fig. S7c was performed once.

**Reporting summary.** Further information on research design is available in the Nature Research Reporting Summary linked to this article.

## Data availability

The mass spectrometry data are available via ProteomeXchange with identifier PXD032072. Source data are provided with this paper. Data used to generate plots and uncropped Western blots and gels are shown in the source data file. All other data supporting the findings of this study are available from the corresponding author upon reasonable request. Source data are provided with this paper.

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

## Acknowledgements

We thank Dr. Xiaochen Wang (Institute of Biophysics, CAS, Beijing, China) for human TAPP1 and 2 ×Fyve plasmids and for help with yeast two-hybrid assay; Dr. Hong Zhang (Institute of Biophysics, CAS, Beijing, China) for human EEA1 and pET-MBP-3C plasmids; Dr. Dong Li (Institute of Biophysics, CAS, Beijing, China) for human RAB5A and RAB7A plasmids; Dr. Robert Kay (MRC Laboratory of Molecular Biology, London, UK) for pDM vectors and Ax2 cells; Dr. Richard Firtel (University of California San

Diego, San Diego, USA) for $pi3k1^-2^-$ cells; and Dr. Li Yu (Tsinghua University, Beijing, China) for HEK 293T cells. We thank the proteomics core facility in Tsinghua University for MS analysis. We also thank the Center for Biological Imaging at the Institute of Biophysics for assistance with data collection. This work was supported by grants from the Ministry of Science and Technology of China (2021YFA1300301 to H.C.), the Strategic Priority Research Program of CAS (XDB37020304 to H.C.), and the National Natural Science Foundation of China (31770894 and 32170701 to H.C. and 31872828 to Y. Yang).

## Author contributions

Author contributions: H.C. and H.T. designed research, analyzed data, and wrote the manuscript; H.T., Z.W., Y.Yuan, X.M., D.L., H.G., and Y.Yang collected the data.

## Competing interests

The authors declare no competing interests.
