## [Peer Review File · Nature Communications]

The PripA-TbcrA complex-centered Rab GAP cascade facilitates macropinosome maturation in DictyosteliumReviewers' Comments:

Reviewer #1:

Remarks to the Author:

The manuscript of Tu et al. identifies PripA-TbcrA as a novel complex that facilitates the Rab5-to-Rab7 switch during macropinosome maturation. The authors provide evidence that mature Rab7-positive macropinosomes fuse with newly formed Rab5-positive macropinosomes during macropinocytosis. Furthermore, the authors describe dynamic localization of the PH domain-containing protein PripA on macropinosomes by interaction with PI(3,4)P2 and Rab7A. They further provide evidence that PripA forms a complex with the GAP-domain containing protein TbcrA that in turn interacts with the active form of Rab5A.

This is an interesting and in parts thorough story, which is mostly based on fluorescence microscopy to track macropinosome maturation. The authors clearly distinguish between different stages of macropinosomes by using the fluid-phase tracer TRITC-dextran and the self-quenching dye DQ-BSA. Furthermore, they convincingly show the interaction of PripA with PI(3,4)P2 and Rab7A as well as TbcrA with Rab5A in various biochemical and cell biological assays. However, the authors lack direct proof for GAP activity of TbcrA towards Rab5a, which is essential to define PripA-TbcrA as a complex that facilitates macropinosome maturation. In addition, the authors utilize inconsistent expression methods for visualization of proteins by fluorescence microscopy. My specific points are listed below:

1. The authors introduce the function of the Mon1-Ccz1 GEF complex: "Mon1 displaces Rabex-5, whereas Ccz1 acts as a GEF for Rab7, leading to Rab7 activation." (line 60-61). However, Mon1-Ccz1 was identified as the GEF complex for Ypt7 in yeast (Nordmann et al., 2010) and Rab7 in human cells (Gerondopoulos et al., 2012). In addition, Mon1-Ccz1 was described as a Rab5 effector (Kinchen et al., 2010; Cui et al., 2014), whereas Mon1/Sand1 alone can displace the Rab5 GEF Rabex5 from membranes in metazoa (Poteryaev et al., 2010). The authors should introduce the function of Mon1-Ccz1 more precisely.
2. The authors argue that overexpression of Rab5 results in enlarged endosomes and defects in endosome maturation. Thus, they express GFP-Rab5A as a single copy from an expression cassette integrated into the genome instead of expression of the GFP fusion protein from an extrachromosomal vector (e.g. Figure 1a). However, the authors do not consistently use the genome-integrated cassette for expression of e.g. Rab7, PripA or TbcrA and instead overexpress these proteins from extrachromosomal vectors (e.g. Figure 1b, 2b, 5b). The authors should utilize the same method for expression of proteins or at least show that expression of Rab7, PripA and TbcrA from extrachromosomal vectors does not affect the localization of proteins or cause cellular defects.
3. Figure 1e shows a pulse-chase experiment, which proves that newly formed Rab5A-marked macropinosomes are first surrounded by Rab7A-positive vesicles containing bright DQ-BSA signal, before they acquire a luminal fluorescent signal. The authors conclude fusion of early Rab5A-positive macropinosomes with mature Rab7A-positive macropinosomes. However, the authors should also provide evidence that the observed Rab7A-positive vesicles are indeed macropinosomes by uptake of TRITC-dextran, which is considered to be specific for macropinocytosis-derived vesicles (as they show in Figure 1b/c). Furthermore, the authors should depict images of the single fluorescence channels instead of exclusively showing the merged images.
4. Figure 4 provides detailed insights into the dynamic localization of PripA during macropinosome maturation. The authors find that PripA co-localizes with Rab5A on newly formed macropinosomes and PripA- and Rab7A-positive vesicles accumulate around these macropinosomes at later stages. However, the authors show different time points for the different stages of macropinosome maturation in Figure 4d and 4e (e.g. 126 vs. 69 sec for the enrichment of PripA or Rab7A around newly formed macropinosomes). The authors should either depict macropinosome maturation at similar time points or alternatively use three-color imaging to follow the localization of Rab5, Rab7 and PripA simultaneously.
5. The authors clearly show the interaction of TbcrA with the active Rab5AQ68L (Figure 5f-h) but they do not provide a direct proof for GAP activity of TbcrA towards Rab5. The evidence that Rab5A from pripA- and tbcrA- cells shows stronger binding to the Rab5 effector EEA1 is rather weak and indirect

(Figure 6e-f). The authors should show GAP activity of Tbcra towards Rab5 in in vitro GAP assays. Therefore, they should utilize an alternative expression system if the required proteins cannot be sufficiently purified from bacteria or Dictyostelium. Furthermore, the authors find that co-expression of PripA with the GAP-dead TbcraR987A but not with the wildtype Tbcra leads to an accumulation of enlarged Rab5-macropinosomes (Figure S5J). From this, they conclude the importance of GAP activity of Tbcra for macropinosome maturation. Here, they should quantify the size of macropinosomes to show the significance of this rather mild phenotype. In addition, other GAPs should be expressed as a control to demonstrate specificity.

Minor issues:

1. The authors should consistently label nascent macropinosomes with asterisks in images of time-lapse microscopy experiments (e.g. missing in Figure 3b).
2. In Figure 3c the authors provide evidence for abolished localization of PripA on newly formed macropinosomes in pi3k1-2- cells. Furthermore, they should include the localization of the PI(3,4)P2 sensor TAPP1, which should also be impaired. This would strengthen the finding that PripA binds PI(3,4)P2.
3. How do the authors explain the absence of PI(3,4)P2 on mature macropinosomes (Figure 7)? In addition, the turnover from Rab5- to Rab7-positive macropinosomes probably occurs more gradually than depicted in panel II of their model (green to red color code). The authors should revise the model and discuss these points in more detail.

Reviewer #2:

Remarks to the Author:

In this MS, the authors discovered a new protein complex, PripA/Tbcra, that regulates the transition of Rab5 and Rab7 during the maturation process of macropinosomes in Dictyostelium. Rab5 and Rab7 are the best characterized Rab proteins in the endocytic process and the phagosome maturation in various eukaryotic organisms. It is not clear about the mechanisms that control the transition from an early Rab5-positive endosome or a phagosome to a late Rab7-positive endosome or a phagosome. Many effector proteins interacting with Rabs and their GEFs and GAPs that regulate their functions have been described. This study shows a novel protein complex acting as RabGAP to regulate the transition of Rab5 and Rab7 during the maturation of macropinosome. They first showed that Rab5 to Rab7 transition occurred during the maturation of macropinosome in Dictyostelium, which is like a Rab transition observed during the maturation of endosome or phagosome in other organisms. They discovered a novel protein, PripA, localized on the macropinosome, and PripA-PH, the N-terminal part, localized in the early macropinosomes and PripA-ΔPH, the C-terminal part, associated with late macropinosomes. They showed that PripA interacted with PI(3,4)P2 via the N-terminal part (PripA-PH) and with active form Rab7A via the C-terminal portion (PripA-ΔPH). They identified a new protein, Tbcra, associating with PripA, using mass spec analysis, and found that portions of PripA contain sequences similar to human RabGAP, TBC1D2, which localizes on late endosome. They showed that Tbcra co-localized with PripA on the macropinosomes. They showed that PripA via the C-terminal part interacts with Tbcra (via the N-terminal region) to form a complex that was recruited on the macropinosome during the Rab5 to Rab7 transition process. Using yeast two-hybrid assay and GST-pulldown assay, they showed that the C-terminal part of Tbcra (Tbcra-TBC) interacted specifically with Rab5 among 26 different Rab proteins. In addition, Tbcra-TBC interacted with the constitutively active form of Rab5A but not the dominant-negative form of Rab5A. They generated pripA, tbcra null cells and found that these mutant cells displayed a modest defect in minimal medium but not in rich liquid medium or on the bacterial lawn, suggesting that the maturation process of micropinocytosis or phagocytosis were not blocked in these mutant cells. They examined the maturation process in the mutant cells and found that the maturation of micropinosome showed a delay in mutant cells. They showed that Rab5 effector (EEA1) pulled down a more Rab5A in pripA or tbcra null cells, suggesting that there were more active Rab5A proteins in the mutants due to a defect in Rab5A inactivation.

Finally, they showed that expressing TbcA but not TbcAR987A (a GAP mutation) rescued the defect of macropinosome maturation in tbcA null cells, arguing that TbcA modulates the maturation process by inactivating Rab5A protein.

This MS contains many interesting results and new findings. It has the potential to be an excellent paper for Nature communication. However, the current version has obvious weaknesses that should be addressed in the revision. I have the following suggestions for the revision:

1: The writing is not clear and precise and needs to be significantly improved. The significance of the work to the field has not been presented clearly in writing. For example, The PCCs for pripA... (on page 11). I do not understand the paragraph. I would like to suggest that the authors work on the entire MS carefully.

2: In Figure 1, Images of more cells and quantification need to be presented to show the transition of Rab5 to Rab7 during micropinocytosis. I suggest that the authors also present multiple cell images and quantification (if possible) in other figures. Such as Figure 3c, which is not clear.

3: Regarding the GAP activity assay, the author indicated that they were not able to purify enough PripA and TbcA for the in vitro assay. Do they need both PripA and TbcA for the assay? Can they simply purify TbcA-TBC to exam the GAP activity in vitro?

4: The data for the transition of Rab5 to Rab7 activation/inactivation is required for the maturation of macropinosome or phagosome is not clear in Dictyostelium. It would be good to compare the maturation process in wild type and the cells expressing CA and DN Rab5 and Rab7.

5: On page 11, the following sentences are not clear. "The PCCs for ...for VacA". "Mutation in the conserved glutamine residue...GAP".

6: PripA and TbcA null cells displayed a modest phenotype in cell growth in liquid medium but not on the bacterial lawn. Could the authors exam the doubling time of these mutant cells using bacterial as food? It is possible that other GAPs controlling Rab5 inactivation in addition to PripA and TbcA.

7: What is the cellular localization of TbcA in pripA null cells?

8: In the introduction and discussion, the author can discuss our current knowledge about the transition of Rab5 to Rab7 in the maturation process of endosome and phagosome in Dictyostelium.

9: It would be nice if they can discuss the role of PripA/TbcA in the maturation of endosome and phagosome.

Reviewer #3:

Remarks to the Author:

Maturation along the endocytic pathway is critical for compartments to acquire their degradative and microbicidal functions. Critically, this involves the transition of GTPases associated with endosomes from Rab5-GTP to Rab7-GTP as seen in all forms of endocytosis including phagocytosis and macropinocytosis. In the past, this transition had been tacitly attributed to the recruitment of Mon1 to displace Rabex-5 (a Rab5 GEF) and Ccz1 to activate Rab7. Mon1 and Ccz1 form a complex once thought to suffice for the transition: The field assumed that the intrinsic hydrolysis of GTP by Rab5 would result in its inactivation without an ongoing exchange for GTP. This was an overly simplistic view and reports for Rab5 GAPs began to surface at least as far back as 2005. There are more recent studies indentifying Rab5 GAPs that feature in the endocytic traffic of yeast (Msb3) and in *C. elegans* (TBC-2). The functional consequences for the loss of these GAPs had been documented, which generally cause gross swelling of endocytic compartments. The mechanisms underlying the spatiotemporal control of their activities, on the other hand, are less well understood, though TBC-2 contains a PH domain and so phosphoinositides had been implicated.

The manuscript by Tu et al. identifies a molecular complex that includes a Rab5-GAP to facilitate this important switch in the maturation of macropinosomes formed in Dictyostelium. Tu et al. describe an adaptor, which they name for its PI-binding and control of Rab5 (PripA), that recruits a Rab5 GAP (TbcA, a predicted RabGAP with some homology to TBC-2 and RabGAP5) to facilitate the Rab5-Rab7 transition. Unlike TBC-2, TbcA does not contain a Pleckstrin Homology (PH) domain. The recruitment

of Tbcra therefore requires an intermediary in the form of PripA, which does contain a PH domain. Interestingly, the authors find that the PH domain of PripA detects PI(3,4)P2 by a protein-lipid overlay assay. The PH domain of TBC-2 had been reported to bind PI3P and PI4P.

The manuscript is well-written, the flow of information is clear, and the experimentation is nicely controlled and executed. In addition, the authors have found a novel PI(3,4)P2 effector, which could be used by the field as a probe in other systems should it prove to be specific. There are some additional experiments required to say this is the case (see below).

The functional experiments describing a role for the Tbcra complex in maturation of the macropinosome, however, are entirely based on its luminal proteolytic activity and these results largely reveal minor consequences to inactivating/deleting the complex which is inconsistent with reports for other Rab5 GAPs. In this model system, the less obvious phenotype could be attributed to functional redundancy from other Rab5 GAPs, or the Rab5-Rab7 transition as being controlled in multiple overlapping ways in Dicty. But it is confusing that the macropinosomes tend to still shrink and distill their contents, as demonstrated by the increased fluorescence of TMR-dextran, given what had been shown in the literature for other Rab5 GAPs. The sustained activity of Rab5 on the nascent macropinosome, should it indeed occur, would arrest traffic, impact acidification of the compartment, etc.

Taking this into account, I propose some experiments below that may help clarify the findings by the authors and perhaps broaden their applicability. I also ask for experiments that determine if the interaction between PripA and PI(3,4)P2 occurs in vivo and for liposomes.

Major comments

1) The PH domain of PripA is proposed to bind to PI(3,4)P2 in vivo based on the use of a lipid strip protein overlay experiment, the colocalization with TAPP1, and the use of mutant cells lacking PI3 kinases; PI3K(1-2)-. The PI3K(1-2)- cells do not just lack PI(3,4)P2 but have 20% of the PIP3 levels as compared to their wildtype counterparts, their mean generation time is 5 times longer, and their macropinocytosis rate is virtually ablated. Yet, the authors are able to show a forming macropinosome (a single video) that does not acquire PripA. More experiments are necessary to specifically say that PripA binds to PI(3,4)P2. I would propose 2 different experiments:

- First, a flotation liposome-based experiment using PI(3,4)P2 versus the monophosphorylated PIs shown to bind other Rab5 GAPs i.e. PI(3)P and PI(4)P as well as dually phosphorylated PI(3,5)P2 and PI(4,5)P2 should be done.
- Second, Gerry Hammond recently published and made available a PI4 phosphatase (INPP4) that can be recruited with rapamycin to membranes of choice or the the plasma membrane constitutively. The would be an excellent tool to more precisely deplete PI(3,4)P2. If this cannot be engineered in the Dicty, I would like to see it be used in the HT1080 system that the authors show in their supplementary files together with the PripA-PH. Since the PI(3,4)P2 probes many of us use are tandem PH domains, and the specificity of the TAPP1 probe has been questioned, PripA-PH-GFP may serve as a new probe for the field.

2) It remains surprising that the loss of perhaps the major Rab5 GAP does not cause swelling of the endosomes nor the traffic of fluid to the lysosome but instead causes the loss of cathepsin activity, albeit to a small extent. The authors do not propose that other Rab5 GAPs operate in the model organism nor do they propose why the cathepsins would be less active. Presumably, this could be due to their not being delivery vectorially to the macropinolysosome or due to changes in the acidification of the compartment. I would ask that this be more formally documented in the following ways:

- Measure the size of early endosomes as was done in their supplementary files for the Rab5A-REMI cells for their PripA- and Tbcra- strains.
- Normalize the DQ-BSA signal to a far-red dextran. Since the endosomes shrink and concentrate the

dextran, as documented by the authors, this is an internal control for the change in volume without the concomitant increase in cathepsin activity.

- Immunostain for cathepsins to determine their delivery to the macropinosomes.

3) In addition to the possibility that PripA could serve as a probe for PI(3,4)P₂, the impact of this work would increase should the authors be able to apply it to a more tractable system like phagocytosis in Dicty. Since the size of the phagosome will remain constant for the period of acquisition (minutes), the authors could readily determine the localization of PripA and its requirement to recruit TbcA and inactivate Rab5. Here too, the authors could look at the degradation of the cargo without changes in the volume of the compartment.

Minor item

The authors mention that there is low expression of PripA and TbcA when expressed in systems to make protein for biochemical assays (e.g. bacteria). The EEA1 pulldown for Rab5 activity appears to show only a small effect (~20%). The authors could consider the overexpression of PripA/TbcA together with Rab5 in HT1080 cells for this experiment. I assume they may have tried antibodies purported to be specific for active Rab5 without success.

REVIEWER COMMENTS

Reviewer #1 (Remarks to the Author):

The manuscript of Tu et al. identifies PripA-TbcrA as a novel complex that facilitates the Rab5-to-Rab7 switch during macropinosome maturation. The authors provide evidence that mature Rab7-positive macropinosomes fuse with newly formed Rab5-positive macropinosomes during macropinocytosis. Furthermore, the authors describe dynamic localization of the PH domain-containing protein PripA on macropinosomes by interaction with PI(3,4)P₂ and Rab7A. They further provide evidence that PripA forms a complex with the GAP-domain containing protein TbcrA that in turn interacts with the active form of Rab5A.

This is an interesting and in parts thorough story, which is mostly based on fluorescence microscopy to track macropinosome maturation. The authors clearly distinguish between different stages of macropinosomes by using the fluid-phase tracer TRITC-dextran and the self-quenching dye DQ-BSA. Furthermore, they convincingly show the interaction of PripA with PI(3,4)P₂ and Rab7A as well as TbcrA with Rab5A in various biochemical and cell biological assays. However, the authors lack direct proof for GAP activity of TbcrA towards Rab5a, which is essential to define PripA-TbcrA as a complex that facilitates macropinosome maturation. In addition, the authors utilize inconsistent expression methods for visualization of proteins by fluorescence microscopy. My specific points are listed below:

1. The authors introduce the function of the Mon1-Ccz1 GEF complex: “Mon1 displaces Rabex-5, whereas Ccz1 acts as a GEF for Rab7, leading to Rab7 activation.” (line 60-61). However, Mon1-Ccz1 was identified as the GEF complex for Ypt7 in yeast (Nordmann et al., 2010) and Rab7 in human cells (Gerondopoulos et al., 2012). In addition, Mon1-Ccz1 was described as a Rab5 effector (Kinchen et al., 2010; Cui et al., 2014), whereas Mon1/Sand1 alone can displace the Rab5 GEF Rabex5 from membranes in metazoa (Poteryaev et al., 2010). The authors should introduce the function of Mon1-Ccz1 more precisely.

RESPONSE: We thank the reviewer for pointing out the error and have revised the manuscript accordingly (page 4, lines 59-60).

2. The authors argue that overexpression of Rab5 results in enlarged endosomes and defects in

endosome maturation. Thus, they express GFP-Rab5A as a single copy from an expression cassette integrated into the genome instead of expression of the GFP fusion protein from an extrachromosomal vector (e.g. Figure 1a). However, the authors do not consistently use the genome-integrated cassette for expression of e.g. Rab7, PripA or TbcA and instead overexpress these proteins from extrachromosomal vectors (e.g. Figure 1b, 2b, 5b). The authors should utilize the same method for expression of proteins or at least show that expression of Rab7, PripA and TbcA from extrachromosomal vectors does not affect the localization of proteins or cause cellular defects.

RESPONSE: We appreciate the reviewer's suggestion. We performed additional experiments to compare the localization of Rab7, PripA, and TbcA expressed by different methods. In general, compared to expression from genome-integrated cassettes, expression of these proteins from extrachromosomal vectors resulted in higher cytosolic background but did not change the localization pattern. Fig S1c and S1d showed that GFP-Rab7A expressed by these different methods exhibited similar localization on cytoplasmic vesicles. Fig S2a showed that, when both Rab7A and Rab5A were expressed from integrated cassettes, the same Rab5-to-Rab7 conversion sequence was observed. After entering the cell, the GFP-Rab5A^{REMI}-marked macropinosome was quickly surrounded by RFP-Rab7A^{REMI}-marked late macropinosomes, and shortly thereafter the membrane perimeter was converted from Rab5A-positive to Rab7A-positive (compared to Fig 1e). Furthermore, expression of GFP-Rab7A from extrachromosomal vectors did not affect the TD uptake and DQ-BSA degradation activity (Fig S1, h and i). Similarly, PripA and TbcA expressed from genome-integrated cassettes exhibited similar localization patterns to that expressed from extrachromosomal vectors (Fig S3, a and b; Fig S6e; Fig S8f). We added the relevant information in the revised manuscript.

3. Figure 1e shows a pulse-chase experiment, which proves that newly formed Rab5A-marked macropinosomes are first surrounded by Rab7A-positive vesicles containing bright DQ-BSA signal, before they acquire a luminal fluorescent signal. The authors conclude fusion of early Rab5A-positive macropinosomes with mature Rab7A-positive macropinosomes. However, the authors should also provide evidence that the observed Rab7A-positive vesicles are indeed macropinosomes by uptake of TRITC-dextran, which is considered to be specific for

macropinocytosis-derived vesicles (as they show in Figure 1b/c). Furthermore, the authors should depict images of the single fluorescence channels instead of exclusively showing the merged images.

RESPONSE: As suggested by the reviewer, we performed the pulse-chase experiment using TRITC-dextran. As shown in Fig S2c, when a newly formed Rab5A-marked macropinosome, which contained no TRITC-dextran, entered the cell, it was quickly surrounded by smaller vesicles containing bright TRITC-dextran signal. Shortly afterwards, it acquired luminal fluorescent signals, indicating that vesicle fusion had occurred to deliver TRITC-dextran, and Rab5A was concurrently released from the macropinosomal membrane. We revised the manuscript (page 7, lines 146-149) to describe the new result. Together, Fig 1e and S2c support an involvement of vesicle fusion during macropinosome maturation. As suggested by the reviewer, we present images of single fluorescence channels as well as merged images in the revised manuscript.

4. Figure 4 provides detailed insights into the dynamic localization of PripA during macropinosome maturation. The authors find that PripA co-localizes with Rab5A on newly formed macropinosomes and PripA- and Rab7A-positive vesicles accumulate around these macropinosomes at later stages. However, the authors show different time points for the different stages of macropinosome maturation in Figure 4d and 4e (e.g. 126 vs. 69 sec for the enrichment of PripA or Rab7A around newly formed macropinosomes). The authors should either depict macropinosome maturation at similar time points or alternatively use three-color imaging to follow the localization of Rab5, Rab7 and PripA simultaneously.

RESPONSE: As suggested by the reviewer, we depict macropinosome maturation at similar time points in Fig 5d and 5e (original Fig 4d and 4e). The entire sequences of events are also shown as supplementary videos 4 and 5. Furthermore, we relabel all time-lapse images of macropinocytosis by defining the frame of cup closure as time "0" so that time courses in different cells can be better compared.

5. The authors clearly show the interaction of Tbc1A with the active Rab5A^{Q68L} (Figure 5f-h) but they do not provide a direct proof for GAP activity of Tbc1A towards Rab5. The evidence

that Rab5A from *pripA*⁻ and *tbcA*⁻ cells shows stronger binding to the Rab5 effector EEA1 is rather weak and indirect (Figure 6e-f). The authors should show GAP activity of TbcA towards Rab5 in in vitro GAP assays. Therefore, they should utilize an alternative expression system if the required proteins cannot be sufficiently purified from bacteria or *Dictyostelium*. Furthermore, the authors find that co-expression of PripA with the GAP-dead TbcAR987A but not with the wildtype TbcA leads to an accumulation of enlarged Rab5-macropinosomes (Figure S5J). From this, they conclude the importance of GAP activity of TbcA for macropinosome maturation. Here, they should quantify the size of macropinosomes to show the significance of this rather mild phenotype. In addition, other GAPs should be expressed as a control to demonstrate specificity.

RESPONSE: We agree with the reviewer that a direct proof for the GAP activity of TbcA towards Rab5 would strengthen this paper. We have been trying to purify full-length TbcA and PripA using different expression systems for several months but have not been successful. To partially address this problem, we purified the TBC domain of TbcA from bacteria using a codon optimized expression construct and examined its GAP activity in vitro. The TBC domain accelerated GTP hydrolysis by Rab5A in a concentration-dependent manner, and mutation of the conserved catalytic residues reduced the GAP activity (Fig 7e). Combined with data showing that Rab5 inactivation was impaired in *tbcA*⁻ and *pripA*⁻ cells and that GAP activity was required for rescuing the DQ-BSA degradation defect in *tbcA*⁻ cells (Fig 7, f-j; Fig 8, e-g), these experiments support the role of the PripA-TbcA complex in promoting Rab5 inactivation during macropinocytosis. We revised the manuscript to incorporate the added information (pages 12-13, lines 292-319).

In this revision, we added data to show that the PripA-TbcA complex also regulates phagosome maturation. PripA and TbcA were recruited to phagosomes (Fig S9, b and c). Although their deletion did not impair phagocytosis-dependent cell growth, mutant cells exhibited compromised Rab5 inactivation and delayed phagocytic cargo degradation (Fig S10). These results provide additional evidence for the GAP function of the complex. The relevant information is included in the revised manuscript (page 15, lines 362-372).

The exact mechanism of how the GAP activity of the PripA-TbcA complex is controlled requires further investigation. Based on our findings, we speculate that TbcA likely functions in

a complex with PripA and Rab7 to ensure that Rab5 is not turned off until the PripA-TbcrA complex responds to Rab7-GTP. This may explain why the activity of purified TBC domain appeared low in vitro. To prove this model will require purification of full-length PripA and TbcrA and comparison of the GAP activity of the complex in the presence and absence of activated Rab7, which we hope can be the subject of future studies.

Regarding the experiment presented in the original Fig S5j, we added additional control and quantified the phenotype as suggested by the reviewer. Because none of the other Rab GAP proteins in *Dictyostelium* have been characterized, it was difficult to determine which one would be the best specificity control. We chose a putative homolog (DDB_G0269982) of yeast Gyp1 protein as the control. Co-expression of PripA and TbcrA^{R987A}, but not WT TbcrA or DDB_G0269982^{R322A}, resulted in the accumulation of macropinosomes marked by GFP-Rab5A. This data is added in the revised manuscript (Fig 8h, 8i, and S8g; page 15, lines 356-358).

Minor issues:

1. The authors should consistently label nascent macropinosomes with asterisks in images of time-lapse microscopy experiments (e.g. missing in Figure 3b).

RESPONSE: As suggested by the reviewer, we used asterisks to label nascent macropinosomes in time-lapse images, except Fig 3g and S3e, where dashed boxes were used to more clearly mark the small macropinosomes formed in *pi3k1-2⁻* cells.

2. In Figure 3c the authors provide evidence for abolished localization of PripA on newly formed macropinosomes in *pi3k1-2⁻* cells. Furthermore, they should include the localization of the PI(3,4)P₂ sensor TAPP1, which should also be impaired. This would strengthen the finding that PripA binds PI(3,4)P₂.

RESPONSE: As suggested by the reviewer, we included TAPP1 in the experiment. As shown in Fig S3e, the localization of PripA and TAPP1 on newly formed macropinosomes was abolished in *pi3k1-2⁻* cells. We also further probed the lipid binding specificity of PripA and found that the PH domain bound specifically to PI(3,4)P₂-coated beads and PI(3,4)P₂-containing liposomes (Fig 3, b and c). In addition, by imaging the membrane association kinetics of PripA-PH and TAPP1, together with sensors for PIP₃ and PI3P, we found that both PripA-PH and TAPP1 were recruited

to macropinosomes after the PIP₃ sensor but prior to the PI3P sensor (Fig 3e, 3f, and S3c). Combined, these experiments strengthen the conclusion that PripA localizes to nascent macropinosomes by binding to PI(3,4)P₂ via the PH domain. We revised the manuscript to incorporate the new data (pages 8-9, lines 182-197). Please see also our response to Reviewer #3, comment # 1.

3. How do the authors explain the absence of PI(3,4)P₂ on mature macropinosomes (Figure 7)? In addition, the turnover from Rab5- to Rab7-positive macropinosomes probably occurs more gradually than depicted in panel II of their model (green to red color code). The authors should revise the model and discuss these points in more detail.

RESPONSE: We thank the reviewer for the helpful suggestions. As discussed above, we observed a sequential accumulation of PI(3,4)P₂ and PI3P on newly generated macropinosomes. A similar sequence of PIP conversion was seen in mammalian cells (Maekawa et al., PNAS, 2014; Welliver et al., Biol Open, 2012). Thus, PI(3,4)P₂ is likely converted to PI3P on more matured macropinosomes. We revised the model to better illustrate the gradual conversion of Rab proteins and PIPs (Fig 9).

Reviewer #2 (Remarks to the Author):

In this MS, the authors discovered a new protein complex, PripA/TbcrA, that regulates the transition of Rab5 and Rab7 during the maturation process of macropinosomes in *Dictyostelium*. Rab5 and Rab7 are the best characterized Rab proteins in the endocytic process and the phagosome maturation in various eukaryotic organisms. It is not clear about the mechanisms that control the transition from an early Rab5-positive endosome or a phagosome to a late Rab7-positive endosome or a phagosome. Many effector proteins interacting with Rabs and their GEFs and GAPs that regulate their functions have been described. This study shows a novel protein complex acting as RabGAP to regulate the transition of Rab5 and Rab7 during the maturation of macropinosome. They first showed that Rab5 to Rab7 transition occurred during the maturation of macropinosome in *Dictyostelium*, which is like a Rab transition observed during the maturation of endosome or phagosome in other organisms. They discovered a novel protein,

PripA, localized on the macropinosome, and PripA-PH, the N-terminal part, localized in the early macropinosomes and Prip- Δ PH, the C-terminal part, associated with late macropinosomes. They showed that PripA interacted with PI(3,4)P₂ via the N-terminal part (PripA-PH) and with active form Rab7A via the C-terminal portion (Prip- Δ PH). They identified a new protein, TbcA, associating with PripA, using mass spec analysis, and found that portions of PripA contain sequences similar to human RabGAP, TBC1D2, which localizes on late endosome. They showed that TbcA co-localized with PripA on the macropinosomes. They showed that PripA via the C-terminal part interacts with TbcA (via the N-terminal region) to form a complex that was recruited on the macropinosome during the Rab5 to Rab7 transition process. Using yeast two-hybrid assay and GST-pulldown assay, they showed that the C-terminal part of TbcA (TbcA-TBC) interacted specifically with Rab5 among 26 different Rab proteins. In addition, TbcA-TBC interacted with the constitutively active form of Rab5A but not the dominant-negative form of Rab5A. They generated pripA, tbcA null cells and found that these mutant cells displayed a modest defect in minimal medium but not in rich liquid medium or on the bacterial lawn, suggesting that the maturation process of macropinocytosis or phagocytosis were not blocked in these mutant cells. They examined the maturation process in the mutant cells and found that the maturation of macropinosome showed a delay in mutant cells. They showed that Rab5 effector (EEA1) pulled down a more Rab5A in pripA or tbcA null cells, suggesting that there were more active Rab5A proteins in the mutants due to a defect in Rab5A inactivation. Finally, they showed that expressing TbcA but not TbcAR987A (a GAP mutation) rescued the defect of macropinosome maturation in tbcA null cells, arguing that TbcA modulates the maturation process by inactivating Rab5A protein.

This MS contains many interesting results and new findings. It has the potential to be an excellent paper for Nature communication. However, the current version has obvious weaknesses that should be addressed in the revision. I have the following suggestions for the revision:

1: The writing is not clear and precise and needs to be significantly improved. The significance of the work to the field has not been presented clearly in writing. For example, The PCCs for pripA.... (on page 11). I do not understand the paragraph. I would like to suggest that the authors work on the entire MS carefully.

RESPONSE: In the original manuscript, we defined Pearson's correlation coefficient as PCC on page 8 but might not use the word properly in the following paragraphs. We apologize for the confusion and have revised the paragraph on page 11 (lines 259-273) to describe more clearly the colocalization experiments. We also worked on the writing of the manuscript (changes are highlighted in blue). We hope the revised version meets the reviewers' requirements.

2: In Figure 1, Images of more cells and quantification need to be presented to show the transition of Rab5 to Rab7 during micropinocytosis. I suggest that the authors also present multiple cell images and quantification (if possible) in other figures. Such as Figure 3c, which is not clear.

RESPONSE: We thank the reviewer for the suggestions. In the revised manuscript, we add additional figures to depict the transition from Rab5 to Rab7, with both Rab5 and Rab7 expressed from genome-integrated cassettes (Fig S2a). The added figure shows a sequence of events similar to that presented in Fig 1e. As a newly formed GFP-Rab5^{REMI}-marked macropinosome entered the cell, it was quickly surrounded by smaller RFP-Rab7A^{REMI}-marked macropinosomes, and shortly thereafter the membrane perimeter was converted from Rab5-positive to Rab7-positive. We present the quantification of Rab5-to-Rab7 conversion in Fig S2b. For Fig 3c, we replaced it with a new figure (new Fig 3g). Due to reduced production of PIP₃ and PI(3,4)P₂, the *pi3k1-2*⁻ cells are severely defective in macropinocytosis and produce only small macropinosomes (Hoeller et al., JCS, 2013). That is probably why the reviewer found the original figure unclear. We repeated the experiment by adding TRITC-dextran to mark macropinosomes. It is clear from the new figure that PripA was not on nascent macropinosomes formed in *pi3k1-2*⁻ cells.

3: Regarding the GAP activity assay, the author indicated that they were not able to purify enough PripA and TbcA for the in vitro assay. Do they need both PripA and TbcA for the assay? Can they simply purify TbcA-TBC to exam the GAP activity in vitro?

RESPONSE: As suggested by the reviewer, we purified the TBC domain of TbcA to examine GAP activity in vitro. Using an optical assay that continuously monitors the release of the inorganic phosphate resulting from GTP hydrolysis, we found that the TBC domain accelerated

GTP hydrolysis by Rab5A in a concentration-dependent manner (Fig 7e). Mutation of the conserved catalytic residues in the TBC domain to alanine reduced the GAP activity. Together with the original data showing that Rab5 inactivation was impaired in *tbcA*⁻ cells and that GAP activity was required for rescuing the DQ-BSA degradation defect associated with *tbcA* deletion (Fig 7, f-j; Fig 8, e-g), these experiments support that TbcA functions as a GAP for Rab5. In addition, we added data in the revised manuscript to show that Rab5 inactivation and cargo degradation were also impaired in *tbcA*⁻ cells during phagocytosis (Fig S10), which provides additional evidence for the GAP activity of TbcA. The relevant information is added in the revised manuscript (pages 12-13, lines 292-319; page 15, lines 362-372).

In cells, TbcA likely functions in a complex with PripA and Rab7 to ensure that Rab5 is not turned off until the PripA-TbcA complex responds to Rab7-GTP. This may explain why the activity of purified TBC domain appeared low in vitro. To prove this model will require the GAP activity of the complex to be measured in the presence of active Rab7. However, purification of full-length PripA and TbcA have been fraught with difficulty. Thus, the exact molecular mechanism of the Rab GAP cascade remains an open question, which we hope can be the subject of future studies.

4: The data for the transition of Rab5 to Rab7 activation/inactivation is required for the maturation of macropinosome or phagosome is not clear in *Dictyostelium*. It would be good to compare the maturation process in wild type and the cells expressing CA and DN Rab5 and Rab7.

RESPONSE: As suggested by the reviewer, we compared the TD uptake and DQ-BSA degradation activity in WT cells and cells expressing the CA and DN forms of Rab5A or Rab7A (Fig S11). The results are in agreement with other findings in our manuscript and previous studies. Firstly, expression of Rab7A^{CA} didn't affect either process. Secondly, expression of Rab5A^{CA} slightly inhibited DQ-BSA degradation but not TD uptake. Consistently, we showed that delayed Rab5 inactivation caused by disruption of *pripA* or *tbcA* resulted in a specific defect in DQ-BSA degradation (revised manuscript, Fig 8a-d). The relatively mild phenotype associated with Rab5A^{CA} expression could be due to the presence of WT Rab5B in cells. Thirdly, expression of Rab5A^{DN} impaired TD uptake (DQ-BSA degradation was likely indirectly

impaired). This data agrees with a recently published study showing that Rab5 activation plays a critical role in promoting macropinosome sealing and scission (Maxson et al., JCS, 2021). Lastly, expression of Rab7A^{DN} markedly inhibited TD uptake, which is consistent with earlier reports in *Dictyostelium* and other systems (Saeed et al., PLoS Pathog, 2010; Rupper et al., Mol Biol Cell, 2001), but the exact mechanism underlying this defect is not fully understood. We added the relevant information and discussion in the revised manuscript (page 16, lines 393-404).

5: On page 11, the following sentences are not clear. “The PCCs for ...for VacA”. “Mutation in the conserved glutamine residue...GAP”.

RESPONSE: We have revised these sentences. Please see the last paragraph on page 11 and lines 281-282 on page 12.

6: PripA and Tbcra null cells displayed a modest phenotype in cell growth in liquid medium but not on the bacterial lawn. Could the authors exam the doubling time of these mutant cells using bacterial as food? It is possible that other GAPs controlling Rab5 inactivation in addition to PripA and Tbcra.

RESPONSE: We measured the doubling time of *pripA*⁻ and *tbcra*⁻ cells using bacteria as food and the speed of bacteria killing in these cells. As shown in Fig S10b-d, deletion of *pripA* or *tbcra* prolonged the survival of GFP-expressing *E.coli* after engulfment (bacterial cell permeabilization and death inferred from the quenching of GFP fluorescence) but did not affect the doubling time of the mutant cells cultured in buffer supplemented with live bacteria.

Combined with data from macropinocytosis, these modest phenotypes indicate that macropinosome maturation and phagosome maturation are not completely blocked in the mutant cells. This may result from the intrinsic activity of Rab5 or yet unknown redundancy in the pathway. Our yeast two hybrid assay (page 12, lines 288-290) may not cover all proteins containing the Rab GAP domain. So at this stage we cannot rule out the possibility that other GAPs control Rab5 inactivation in addition to Tbcra. We discussed this point in the revised manuscript (page 18, lines 440-443).

7: What is the cellular localization of Tbcra in pripA null cells?

RESPONSE: Prompted by the reviewer, we examined the localization of GFP-TbcrA in *pripA*⁻ cells. As shown in Fig S6e and S9c, deletion of *pripA* greatly impaired the recruitment of TbcrA to macropinosomes and phagosomes. These observations are consistent with the findings showing that PripA and TbcrA interact with each other and their knockout cells exhibit similar defects. However, the recruitment of TbcrA was not abolished by *pripA* deletion, as quantified in Fig S9d, suggesting that additional signals may contribute to this process. We speculate that one possible candidate that may also regulate the localization of TbcrA is Rab5 and would like to validate this in future studies. We added the relevant information and discussion in the revised manuscript (page 13, lines 315-316; page 15, lines 367-368; page 17, lines 413-414).

8: In the introduction and discussion, the author can discuss our current knowledge about the transition of Rab5 to Rab7 in the maturation process of endosome and phagosome in *Dictyostelium*.

RESPONSE: We thank the reviewer for the suggestion. Previous studies on phagocytosis in *Dictyostelium* during *Mycobacterium marinum* infection revealed that Rab5 was recruited to newly formed phagosomes and was withdrawn approximately ten minutes later (Barisch et al., Methods Mol Biol, 2015), whereas Rab7A was detected on phagosomes at a later stage (Cardenal-Muñoz et al., PLoS Pathog, 2017). We added the relevant information in the revised manuscript (page 15, lines 362-363). However, to the best of our knowledge, there has not been a study showing the transition of Rab5 to Rab7 during endosome or phagosome maturation in *Dictyostelium* by live cell imaging.

9: It would be nice if they can discuss the role of PripA/TbcrA in the maturation of endosome and phagosome.

RESPONSE: During revision, we performed additional experiments to investigate the function of the PripA-TbcrA complex during phagocytosis. We found that the transition of Rab5 to Rab7 on phagosomal membranes appeared to follow a sequence of events similar to that observed during macropinocytosis (Fig S9a and video 7). PripA and TbcrA were also recruited to phagosomes (Fig S9, b and c; video 8). Although deletion of *pripA* or *tbcra* did not affect bacterial phagocytosis-dependent cell growth as discussed above, the mutant cells exhibited

compromised Rab5 inactivation and delayed phagocytic cargo degradation (Fig S10). Therefore, the PripA/TbcrA-centered Rab GAP cascade appears to also regulate phagosome maturation. The relevant information is added in the revised manuscript (page 15, lines 362-372).

Reviewer #3 (Remarks to the Author):

Maturation along the endocytic pathway is critical for compartments to acquire their degradative and microbicidal functions. Critically, this involves the transition of GTPases associated with endosomes from Rab5-GTP to Rab7-GTP as seen in all forms of endocytosis including phagocytosis and macropinocytosis. In the past, this transition had been tacitly attributed to the recruitment of Mon1 to displace Rabex-5 (a Rab5 GEF) and Ccz1 to activate Rab7. Mon1 and Ccz1 form a complex once thought to suffice for the transition: The field assumed that the intrinsic hydrolysis of GTP by Rab5 would result in its inactivation without an ongoing exchange for GTP. This was an overly simplistic view and reports for Rab5 GAPs began to surface at least as far back as 2005. There are more recent studies indentifying Rab5 GAPs that feature in the endocytic traffic of yeast (Msb3) and in *C. elegans* (TBC-2). The functional consequences for the loss of these GAPs had been documented, which generally cause gross swelling of endocytic compartments. The mechanisms underlying the spatiotemporal control of their activities, on the other hand, are less well understood, though TBC-2 contains a PH domain and so phosphoinositides had been implicated.

The manuscript by Tu et al. identifies a molecular complex that includes a Rab5-GAP to facilitate this important switch in the maturation of macropinosomes formed in *Dictyostelium*. Tu et al. describe an adaptor, which they name for its PI-binding and control of Rab5 (PripA), that recruits a Rab5 GAP (TbcrA, a predicted RabGAP with some homology to TBC-2 and RabGAP5) to facilitate the Rab5-Rab7 transition. Unlike TBC-2, TbcrA does not contain a Pleckstrin Homology (PH) domain. The recruitment of TbcrA therefore requires an intermediary in the form of PripA, which does contain a PH domain. Interestingly, the authors find that the PH domain of PripA detects PI(3,4)P₂ by a protein-lipid overlay assay. The PH domain of TBC-2 had been reported to bind PI3P and PI4P.

The manuscript is well-written, the flow of information is clear, and the experimentation is nicely controlled and executed. In addition, the authors have found a novel PI(3,4)P₂ effector, which could be used by the field as a probe in other systems should it prove to be specific. There are some additional experiments required to say this is the case (see below).

The functional experiments describing a role for the TbcA complex in maturation of the macropinosome, however, are entirely based on its luminal proteolytic activity and these results largely reveal minor consequences to inactivating/deleting the complex which is inconsistent with reports for other Rab5 GAPs. In this model system, the less obvious phenotype could be attributed to functional redundancy from other Rab5 GAPs, or the Rab5-Rab7 transition as being controlled in multiple overlapping ways in Dicty. But it is confusing that the macropinosomes tend to still shrink and distill their contents, as demonstrated by the increased fluorescence of TMR-dextran, given what had been shown in the literature for other Rab5 GAPs. The sustained activity of Rab5 on the nascent macropinosome, should it indeed occur, would arrest traffic, impact acidification of the compartment, etc.

Taking this into account, I propose some experiments below that may help clarify the findings by the authors and perhaps broaden their applicability. I also ask for experiments that determine if the interaction between PripA and PI(3,4)P₂ occurs in vivo and for liposomes.

Major comments

1) The PH domain of PripA is proposed to bind to PI(3,4)P₂ in vivo based on the use of a lipid strip protein overlay experiment, the colocalization with TAPP1, and the use of mutant cells lacking PI3 kinases; PI3K(1-2)⁻. The PI3K(1-2)⁻ cells do not just lack PI(3,4)P₂ but have 20% of the PIP₃ levels as compared to their wildtype counterparts, their mean generation time is 5 times longer, and their macropinocytosis rate is virtually ablated. Yet, the authors are able to show a forming macropinosome (a single video) that does not acquire PripA. More experiments are necessary to specifically say that PripA binds to PI(3,4)P₂. I would propose 2 different experiments:

- First, a flotation liposome-based experiment using PI(3,4)P₂ versus the monophosphorylated PIs shown to bind other Rab5 GAPs i.e. PI(3)P and PI(4)P as well as dually phosphorylated

PI(3,5)P₂ and PI(4,5)P₂ should be done.

- Second, Gerry Hammond recently published and made available a PI4 phosphatase (INPP4) that can be recruited with rapamycin to membranes of choice or the the plasma membrane constitutively. This would be an excellent tool to more precisely deplete PI(3,4)P₂. If this cannot be engineered in the Dicty, I would like to see it be used in the HT1080 system that the authors show in their supplementary files together with the PripA-PH. Since the PI(3,4)P₂ probes many of us use are tandem PH domains, and the specificity of the TAPP1 probe has been questioned, PripA-PH-GFP may serve as a new probe for the field.

RESPONSE: We thank the reviewer for these helpful suggestions and have performed additional experiments to verify the interaction between PripA-PH and PI(3,4)P₂. Firstly, using PIP coated agarose beads and liposome flotation assays, we confirmed that PripA-PH (124-219 aa) bound specifically to PI(3,4)P₂ (Fig 3b and 3c). Secondly, we found that the localization dynamics of PripA-PH during macropinocytosis matched that of the PI(3,4)P₂ sensors. PripA-PH and TAPP1 were recruited to nascent macropinosomes after the PIP₃ sensor GRP1-PH but prior to the PI(3)P sensor 2×Fyve (Fig 3e and 3f; Fig S3c and 3d; our published result in Yang et al., JCB, 2021, Fig 3I). Sequential accumulation of a PIP₃ sensor and the presumably more specific PI(3,4)P₂ sensor cPH×3 was also observed in previous publication (Yang et al., JCB, 2021, Fig S1D). These experiments demonstrate that PIP₃, PI(3,4)P₂, and PI(3)P emerge sequentially during macropinosome maturation, similar to that reported in mammalian cells (Maekawa et al., PNAS, 2014; Welliver et al., Biol Open, 2012). Thirdly, we repeated the experiment presented in the original Fig 3c, which the reviewer found unconvincing, by adding TRITC-dextran to mark macropinosomes and coexpressing PripA and TAPP1. It is clear from new Fig 3g and S3e that PripA and TAPP1 are not present on nascent macropinosomes formed in *pi3k1-2*⁻ cells. We believe the added data support our conclusion that PripA localizes to newly formed macropinosomes by interacting with PI(3,4)P₂ via the PH domain. We revised the manuscript to incorporate the new data (pages 8-9, lines 182-197).

2) It remains surprising that the loss of perhaps the major Rab5 GAP does not cause swelling of the endosomes nor the traffic of fluid to the lysosome but instead causes the loss of cathepsin activity, albeit to a small extent. The authors do not propose that other Rab5 GAPs operate in the

model organism nor do they propose why the cathepsins would be less active. Presumably, this could be due to their not being delivered vectorially to the macropinosome or due to changes in the acidification of the compartment. I would ask that this be more formally documented in the following ways:

- Measure the size of early endosomes as was done in their supplementary files for the Rab5A-REMI cells for their *PripA*- and *TbcrA*- strains.
- Normalize the DQ-BSA signal to a far-red dextran. Since the endosomes shrink and concentrate the dextran, as documented by the authors, this is an internal control for the change in volume without the concomitant increase in cathepsin activity.
- Immunostain for cathepsins to determine their delivery to the macropinosomes.

RESPONSE: We understand the main concern of the reviewer was whether deletion of *pripA* or *tbcra* caused swelling of macropinosomes, which could account for decreased DQ-BSA signal in the mutant cells. To address this concern, we measured the size of macropinosomes as suggested by the reviewer. Deletion of *pripA* or *tbcra* did not evidently change the size of macropinosomes containing TD or DQ-BSA (Fig S8, b and c). The size of EEA1-RFP-labeled macropinosomes also appeared unaffected (Fig 7j). In addition, we found that phagocytic cargo degradation was similarly delayed in the mutant cells (Fig S10, c and d). Together, these experiments indicate that changes in the volume of endocytic compartments are unlikely the cause of defects in cargo degradation.

We speculate that the size of macropinosomes remains largely unaffected for the following reasons. Firstly, there may be yet unknown redundancy in the pathway, which also explains the mild phenotypes seen in the mutant cells. Our Y2H assay (page 12, lines 288-290) may not cover all Rab GAP domain-containing proteins. In addition, the assay condition may not be optimized. A previous study showed that the catalytic arginine residue needs to be mutated in order to see interaction between RabGAP-5 and Rab5 in Y2H (Haas et al., Nat Cell Biol, 2005). Thus, it is possible that other GAPs stimulate Rab5 inactivation in the absence of *TbcrA*. Secondly, it was shown that the main driver of volume change for early macropinosomes is the loss of osmolytes and osmotically-coupled water (Freeman et al., Science, 2019), which may not be affected by delayed Rab5 inactivation. Third, persistent Rab5 activation causes enlargement of early endosomes is mainly because it stimulates homotypic endosome fusion. Such events may not

occur frequently enough during bulk endocytosis. The limited number of nascent macropinosomes or phagosomes in cells could preclude excessive homotypic fusion. In line with this idea, we found that even expression of the CA form of Rab5 did not evidently change the size of early macropinosome. Similarly, judging from data presented in previous studies on phagocytosis, the size of phagosomes was not affected by expression of the CA form of Rab5 or deletion of its GAP (Li et al., Development, 2009).

It had become difficult for us to obtain antibody/serum from a foreign lab since the pandemic, so we were not able to perform the immunostaining experiment suggested by the reviewer. We do think that the cargo degradation defects in the mutants are caused by changes in the composition of macropinosome compartments. We observed frequent fusion events between early and late macropinosomes. This likely delivers components that have already been anchored on or enclosed in late macropinosomes, such as hydrolytic enzymes, to early macropinosomes, thereby promoting cargo processing. A previous study also observed concentrated contents of preexisting macropinosomes being added to new macropinosomes (Clarke et al., Traffic, 2002). The ability of PripA to link early and late macropinosomes and TbcA to stimulate Rab5 inactivation may promote these processes. However, the exact mechanism requires further investigation, which we hope can be the subject of future studies.

3) In addition to the possibility that PripA could serve as a probe for PI(3,4)P₂, the impact of this work would increase should the authors be able to apply it to a more tractable system like phagocytosis in Dicty. Since the size of the phagosome will remain constant for the period of acquisition (minutes), the authors could readily determine the localization of PripA and its requirement to recruit TbcA and inactivate Rab5. Here too, the authors could look at the degradation of the cargo without changes in the volume of the compartment.

RESPONSE: We thank the reviewer for the suggestion. In the original manuscript, we showed that the transition of Rab5 to Rab7 on phagosomal membranes appeared to follow a sequence of events similar to that observed during macropinocytosis (revised manuscript, Fig S9a and video 7). As suggested by the reviewer, we performed additional experiments to investigate the function of the PripA-TbcA complex in phagocytosis. We found that PripA and TbcA were also recruited to phagosomes (Fig S9, b and c; video 8). Deletion of *pripA* impaired the phagosomal

association of TbcA (Fig S9c). Furthermore, although deletion of *pripA* or *tbcA* did not affect bacterial phagocytosis-dependent cell growth, the mutant cells exhibited delayed Rab5 inactivation and phagocytic cargo degradation (Fig S10). Therefore, PripA and TbcA seem to also regulate phagosome maturation. We revised the manuscript to incorporate the new data (page 15, lines 362-372).

Minor item

The authors mention that there is low expression of PripA and TbcA when expressed in systems to make protein for biochemical assays (e.g. bacteria). The EEA1 pulldown for Rab5 activity appears to show only a small effect (~20%). The authors could consider the overexpression of PripA/TbcA together with Rab5 in HT1080 cells for this experiment. I assume they may have tried antibodies purported to be specific for active Rab5 without success.

RESPONSE: We appreciate the reviewer's suggestion. We have attempted to express and purify full-length TbcA and PripA in different systems but have not been successful. To partially address this problem and seek further evidence for the GAP activity of TbcA, we purified the TBC domain from bacteria using a codon optimized construct and examined its GAP activity in vitro. The TBC domain accelerated GTP hydrolysis by Rab5A in a concentration-dependent manner and mutation of the conserved catalytic residues reduced the GAP activity (Fig 7e). Combined with data showing that Rab5 inactivation was impaired in *tbcA*⁻ cells during macropinocytosis and phagocytosis (Fig 7, f-j; Fig S10a), we believe these experiments collectively demonstrate that TbcA functions as a Rab5 GAP (please see also our response to Reviewer #1, comment #5).

Reviewers' Comments:

Reviewer #1:

Remarks to the Author:

The authors addressed all my concerns, in particular the GAP assay (7E). I recommend that they include here also a quantification.

The model in Figure 9 would make more sense if Tbc1A would be in a different state in I and II to show that Rab7 changes something.

Otherwise I am fine with the revision.

Reviewer #2:

Remarks to the Author:

The authors added new experiments and modified the text to address the concerns. The revised MS has been significantly improved. I support its publication in Nature Communications.

Reviewer #3:

Remarks to the Author:

The authors have done a nice job of addressing the reviewer comments and added important experiments that broaden its implications for the field. The notion that PI3,4P2 orchestrates the Rab5-Rab7 switch via their GAP complex is well-supported in their experimental work and the manuscript remains well-written and organized.

I look forward to seeing this published.

REVIEWER COMMENTS

Reviewer #1 (Remarks to the Author):

The authors addressed all my concerns, in particular the GAP assay (7E). I recommend that they include here also a quantification.

RESPONSE: As suggested by the reviewer, we included quantification of the GAP assay. By fitting the data to a Michaelis–Menten model function, we found that the catalytic efficiency of TBC domain ($802.2 \pm 58.5 \text{ M}^{-1}\text{s}^{-1}$) was approximately twofold higher than that of mutated TBC domain ($357.2 \pm 40.3 \text{ M}^{-1}\text{s}^{-1}$). We revised the manuscript to incorporate this information (page 12, lines 298-300).

The model in Figure 9 would make more sense if TbcA would be in a different state in I and II to show that Rab7 changes something.

Otherwise I am fine with the revision.

RESPONSE: We thank the reviewer for the suggestion and have revised the model and figure legend. We explained in the legend that the PripA-TbcA complex may be required to sharpen Rab5 activation at stage I, or alternatively, it may not yet become fully active (indicated by the question mark). During stage II, interactions among active Rab7, PripA, and TbcA may stimulate the GAP activity of TbcA to promote Rab5-GTP hydrolysis. In this way, Rab5 is turned off when the PripA-TbcA complex responds to Rab7-GTP, which ensures consecutive function of Rab5 and Rab7.

Reviewer #2 (Remarks to the Author):

The authors added new experiments and modified the text to address the concerns. The revised MS has been significantly improved. I support its publication in Nature Communications.

Reviewer #3 (Remarks to the Author):

The authors have done a nice job of addressing the reviewer comments and added important

experiments that broaden its implications for the field. The notion that PI(3,4)P₂ orchestrates the Rab5-Rab7 switch via their GAP complex is well-supported in their experimental work and the manuscript remains well-written and organized.

I look forward to seeing this published.